# Mosquito-disseminated diflubenzuron and spinosad as alternatives to mosquito-disseminated pyriproxyfen: a proof-of-concept, blind, controlled comparison in experimental cages

**Gabriela Brandina Aquino de Abreu[1], Ayrton Sena Gouveia[1,2,3], Francisco Augusto da Silva Ferreira[1], José Joaquín Carvajal-Cortés[1], Cláudia Torres Codeço[3], Fernando Abad-Franch[1,4/+], Sérgio Luiz Bessa Luz[1/+]**

[1]Fundação Oswaldo Cruz-Fiocruz, Instituto Leônidas e Maria Deane, Núcleo de Patógenos, Reservatórios e Vetores na Amazônia, Manaus, AM, Brasil
[2]Fundação Oswaldo Cruz-Fiocruz, Instituto Oswaldo Cruz, Programa de Pós-Graduação em Biologia Parasitária, Rio de Janeiro, RJ, Brasil
[3]Fundação Oswaldo Cruz-Fiocruz, Instituto Oswaldo Cruz, Programa de Computação Científica, Rio de Janeiro, RJ, Brasil
[4]Universidade Federal de Minas Gerais, Instituto de Ciências Biológicas, Departamento de Parasitologia, Laboratório de Ecologia de Parasitos e Vetores, Belo Horizonte, MG, Brasil

**BACKGROUND** Mosquito-disseminated pyriproxyfen (MD-PPF) is a promising novel tool for urban-mosquito control, yet resistance to PPF (a juvenile-hormone analogue) may arise in exposed mosquito populations. Alternative larvicide/pupicide molecules suitable for mosquito-driven dissemination, but with distinct modes of action, are therefore needed.

**OBJECTIVES** To provide a proof-of-concept evaluation of mosquito-disseminated diflubenzuron (MD-DFB, a chitin-synthesis inhibitor) and spinosad (MD-SPN, a biological neurotoxin composite) as potential alternatives to MD-PPF.

**METHODS** We studied *Aedes aegypti*-driven dissemination in 20 blind, controlled experiments run in 110 × 90 × 30-cm cages. Of primary interest was whether and how (a) mosquito-driven dissemination affected adult-mosquito emergence (1705 larvae in 40 open and 20 closed cups set inside cages; generalised linear mixed models) and (b) exposure to larvicide/pupicide-treated dissemination stations affected adult-female lifespan (400 females released inside cages; proportional-hazards mixed models).

**FINDINGS** Adult-mosquito emergence was similar across treatments in closed cups. In open cups, average emergence fell from ~90% [95% confidence interval (CI), 84-95%] in control cages to ~30% (20-43%), ~56% (42-69%), and ~75% (63-85%) in, respectively, MD-PPF, MD-DFB, and MD-SPN cages. Exposure to SPN, but not to DFB or PPF, clearly reduced adult-female lifespan (SPN death-hazard ratio 2.4; 1.2-5.0).

**CONCLUSION** Mosquito-disseminated diflubenzuron holds promise as a potential alternative to MD-PPF; further testing in field settings seems warranted.

Key words: mosquito control - *Aedes* - insecticide auto-dissemination - resistance management

Mosquitoes are notorious biting pests and the vectors of many human pathogens including viruses (dengue, Zika, chikungunya, yellow fever and more) and filarial worms.[1,2,3,4] Mosquito species adapted to urban habitats are particularly dangerous, because they can fuel pathogen spread across large human populations.[5,6,7] Urban-mosquito control is, therefore, critical to protecting public health, yet none of the many tactics tested to date has been fully successful.[1-7] One key drawback is that the aquatic habitats in which urban mosquito juveniles (most notably *Aedes aegypti* juveniles) develop are often small, cryptic, and hidden or inaccessible, and therefore remain untreated or unmanaged during vector-control activities; as a result, the fraction of mosquito juvenile aquatic hab-

itats that are effectively treated or managed (*i.e.*, 'breeding-site coverage') is usually low, and this may reduce the effectiveness of mosquito control interventions.[8,9,10]

One promising novel strategy involves using the mosquitoes themselves to disseminate pyriproxyfen (PPF), a potent insect juvenile-hormone analogue, from lure 'dissemination stations' treated with fine PPF powder to otherwise untreated aquatic larval habitats.[11,12,13,14] This strategy, known as 'PPF autodissemination' or 'mosquito-disseminated PPF' (MD-PPF), has been shown in field trials to yield high breeding-site coverage (likely because urban *Aedes* spp. spread their eggs across several sites and prefer artificial containers),[13,14,15] leading to sharp increases of juvenile-mosquito mortality and

Financial support: CAPES (finance code 001), FAPEAM (call 04/2022 - Inovação na Amazônia), Ministério da Saúde, Brasil.
GBAA and AS contributed equally to this work.
+ Corresponding authors: abadfranch@ufmg.br |  https://orcid.org/0000-0002-7715-0328 / sergio.luz@fiocruz.br |  https://orcid.org/0000-0001-5887-7372

sharp declines of adult-mosquito emergence — which, in turn, can lead to lower adult-mosquito densities and reduced dengue incidence.[15,16,17,18] While these and other results[19] suggest that MD-PPF can become a useful addition to the urban-mosquito control toolbox, experience shows that the large-scale, long-term use of insecticides has the potential to, and most likely will, select for resistant mosquitoes.[20,21,22,23]

Insecticide-resistance management depends on the availability of active ingredients with diverse modes of action and that can be deployed safely and conveniently at scale.[22] In the case of mosquito-disseminated insecticides for urban-mosquito control, this translates into the need for chemicals that (i) effectively kill juvenile mosquitoes at the low doses that adult mosquitoes can pick and disseminate; (ii) do not knock-down or quickly kill adult mosquitoes (which would hamper dissemination); (iii) are formulated as powder particles suitable for mosquito-driven dissemination; (iv) have mechanisms of action that differ from that of the insecticide towards which mosquitoes have evolved resistance; and (v) are safe for humans and other vertebrates at the low concentrations expected to derive from mosquito-driven dissemination to mosquito larval habitats — which, in practice, means having been evaluated and approved for use in drinking water by the World Health Organization (WHO).[24,25,26,27]

We conducted a proof-of-concept study to test whether a chitin-synthesis inhibitor (diflubenzuron, DFB) and a biological neurotoxin composite (spinosad, SPN), both WHO-approved and in regular use for mosquito control,[26,27] may be useful for urban-mosquito control strategies based on mosquito-driven dissemination. Using replicate blind, controlled experiments, we evaluated (i) whether *Ae. aegypti* females disseminate PPF, DFB, and SPN from dissemination stations (DSs) to untreated larval habitats inside experimental cages; (ii) the effects of such dissemination on juvenile *Ae. aegypti* development; and (iii) whether and how exposure to insecticide-treated DSs would affect the lifespan and death hazard of adult *Ae. aegypti* females. While we expected SPN to shorten adult-female lifespan, we wanted to test whether, in the specific context of mosquito-driven dissemination, this effect was slow enough to allow for some measurable degree of larvicide spread from DSs. As it turned out, SPN quickly killed many females, and only DFB emerged as a potentially useful alternative or complement to pyriproxyfen for mosquito-driven dissemination.

## MATERIALS AND METHODS

*Mosquitoes* - Using oviposition traps, we collected *Ae. aegypti* eggs from two neighbourhoods of Manaus, Amazonas, Brazil and established a laboratory colony; we then used first-generation offspring from that colony in all our experiments.

*Insecticides* - We used commercially available formulations of PPF (0.5% granules; Sumitomo Chemical, Tokyo, Japan), DFB (25% wettable powder; Champion Farmoquímico Ltd., Anápolis, Brazil), and SPN (7.48% effervescent tablets; Clarke Mosquito Control

Inc., Roselle, USA). PPF granules and SPN tablets were ground to fine, talcum-like powder for use in the experimental procedures detailed below.

*Controls* - We treated PPF, which has been tested extensively for mosquito-driven dissemination,[11-19] as our positive control in all experiments. As a negative control (abbreviated 'CTR' hereafter), all experiments included replicates using a sham 'treatment' or 'placebo' with a non-insecticidal powder — fine pumice powder (Asfer, São Caetano do Sul, Brazil), which is similar in appearance to the insecticide powders we used.

*Blinding* - Experimenters evaluating treatment effects on any outcome (juvenile development and mortality, adult-female lifespan and death hazard) were blinded to the identity of the products used in each replicate — whether PPF, DFB, SPN, or CTR. For this, research team members not involved in outcome evaluation labelled all items (bottles, flasks, DSs, cups, cages, etc. as detailed below) with unique codes to which experimenters involved in outcome evaluation did not have access. Codes were linked back to treatments only after all experiments were completed.

*Diagnostic-concentration bioassays* - We first did standard diagnostic-concentration larval bioassays[28] to test whether our mosquito population was susceptible to PPF, DFB, and SPN. Diagnostic concentrations were defined as approximately twice the 99% lethal concentration ($LC_{99}$) of active ingredient;[28] based on published results, we used 0.02 parts per million (ppm) for PPF, 1.5 ppm for DFB, and 1 ppm for SPN.[29,30,31] We prepared stock solutions by mixing (over 1 h on a shaker) 0.5 L of distilled water with either PPF (5 g of 0.5% powder = 50 ppm); DFB (5 g of 25% powder = 2500 ppm); SPN (6.7 g of 7.48% powder = 1000 ppm); or pumice powder (5 g). Each stock solution was stored in a glass bottle wrapped with aluminium foil and coded with a randomly assigned letter (A, B, C or D). We hatched 800 *Ae. aegypti* eggs from our colony and reared the larvae to stage LIII; we then separated 40 cohorts of 20 larvae each in 200-mL individually-coded plastic cups with 100 mL of tap water and a pinch of fish food. A researcher blinded to stock solution identity then pipetted, using disposable tips, a prescribed amount of each stock solution — 40 μL for stock solution B (containing PPF), 60 μL for stock solution D (with DFB), 100 μL for stock solution A (SPN), and 50 μL for stock solution C (pumice powder) — into each individually-identified cup, and gently stirred the mix using a disposable stick. The cups were then closed with cotton cloth and checked daily (by blinded evaluators) to record dead larvae or pupae (which were removed using disposable plastic pipettes) and emerging adults (removed using a Castro aspirator) until all 20 juveniles in each cup had either died or emerged as adults.

*Mosquito-driven dissemination: cage experiments* - We built ten 110 × 90 × 30-cm mosquito cages (Fig. 1) to run the dissemination experiments; we ran two replication rounds, with cages thoroughly washed between rounds (Fig. 2). We used digital thermo-hygrometers

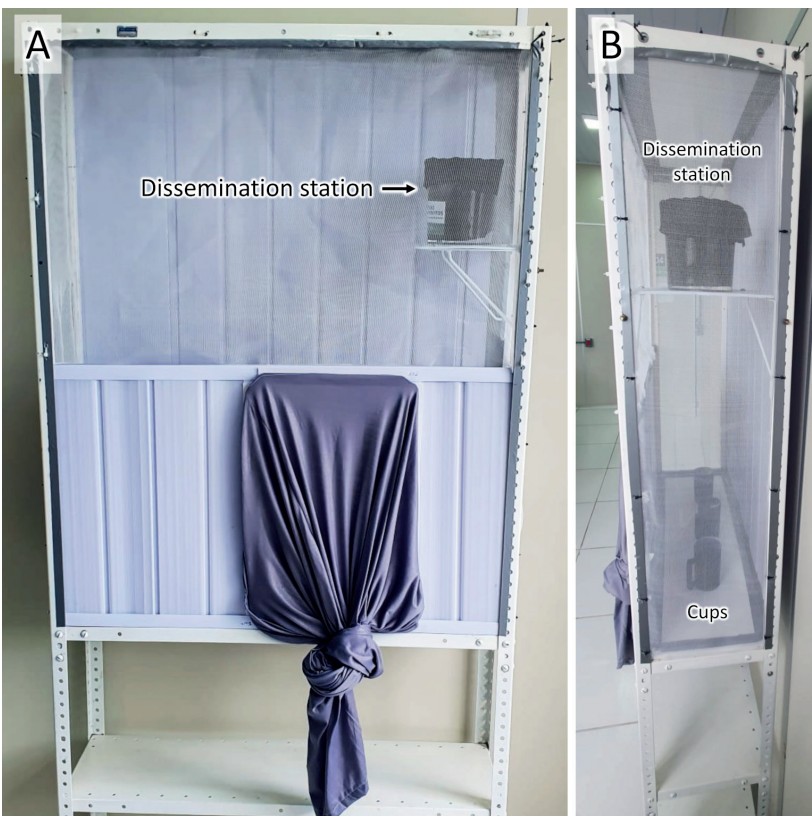

Fig. 1: cages used in mosquito-driven dissemination experiments. Cages (110 × 90 × 30 cm) were built using metal frame, PVC panels, and nylon mesh. (A) frontal view, showing a dissemination station placed on a small raised shelf; (B) lateral view, showing also the three small oviposition cups placed on the bottom of the cage.

(1566-1, J.Prolab, São José dos Pinhais, Brazil) and thermometers (TDU-100, Unity, Guarulhos, Brazil) to record daily mean, maximum, and minimum values of (i) temperature (ºC) in each cage and (ii) relative humidity (RH, %) in two rooms, each housing five cages per round. Within each cage, we placed one DS on a small raised shelf and three oviposition cups on the cage floor (Figs. 1-2); 20 mated and blood-fed, 7-10 days-old *Ae. aegypti* females were then released inside each cage and supplied with water and 10% sucrose solution in separate cotton balls, which were replaced twice weekly. DSs were 2-L plastic cups (Fig. 1) filled with ~1800 mL of tap water and with the walls lined with black cloth dusted with ~5 g/m² of either PPF, DFDB, SPN, or CTR powder.[13,15] Each oviposition cup had 100 mL of tap water with fish food, a strip of filter paper that mosquito females could use for egg laying, and 20 *Ae. aegypti* LIII larvae hatched from our colony (see above). Two of the cups were open, and hence available for contamination with mosquito-disseminated insecticide, whereas the third cup was closed with nonwoven fabric (from disposable lab coats) and was meant to act as an 'in-cage' negative control. We removed DSs and oviposition cups from the cages after seven days. Adult mosquitoes remained inside the cages until death or for up to 60 days.

To measure the possible effects of mosquito-driven dissemination on juvenile-mosquito development, blinded evaluators checked the cups daily as in the bioassays described above. To measure the possible effects of exposure to insecticide powder on adult female lifespan and death hazard, blinded evaluators checked the cages daily for dead specimens.

*Data analysis* - We first describe and summarise our data in tables and graphs; for proportions, we compute score 95% confidence intervals (CI) using *Hmisc* 5.2-3.[32] We then use mixed models including (i) binomial generalised linear mixed models (logit link-function), fitted using *glmmTMB* 1.1.10,[33] for assessing treatment effects on adult-mosquito emergence; and (ii) proportional-hazards (Cox) mixed models, fitted using *coxme* 2.2-22,[34] for assessing treatment effects on adult-mosquito lifespan. These models (a) incorporate fixed effects of the four-level treatment factor (and, in the case of cage experiments, an interaction between treatment and open cup), (b) adjust for covariates where relevant, and (c) explicitly account for the non-independence of observations stemming from the same cup and the same cage, as appropriate, via random intercepts. When more than a single model was fitted to the same data, we used Schwartz's information criterion (also known as the Bayesian information criterion, or BIC)[35] to assess relative model performance, and then focused on the results from the best-performing model; we used *ggeffects* 2.1.0[36] to compute marginal mean predictions and CIs from those focal models. Details on model structure and specification are given in the RESULTS section to

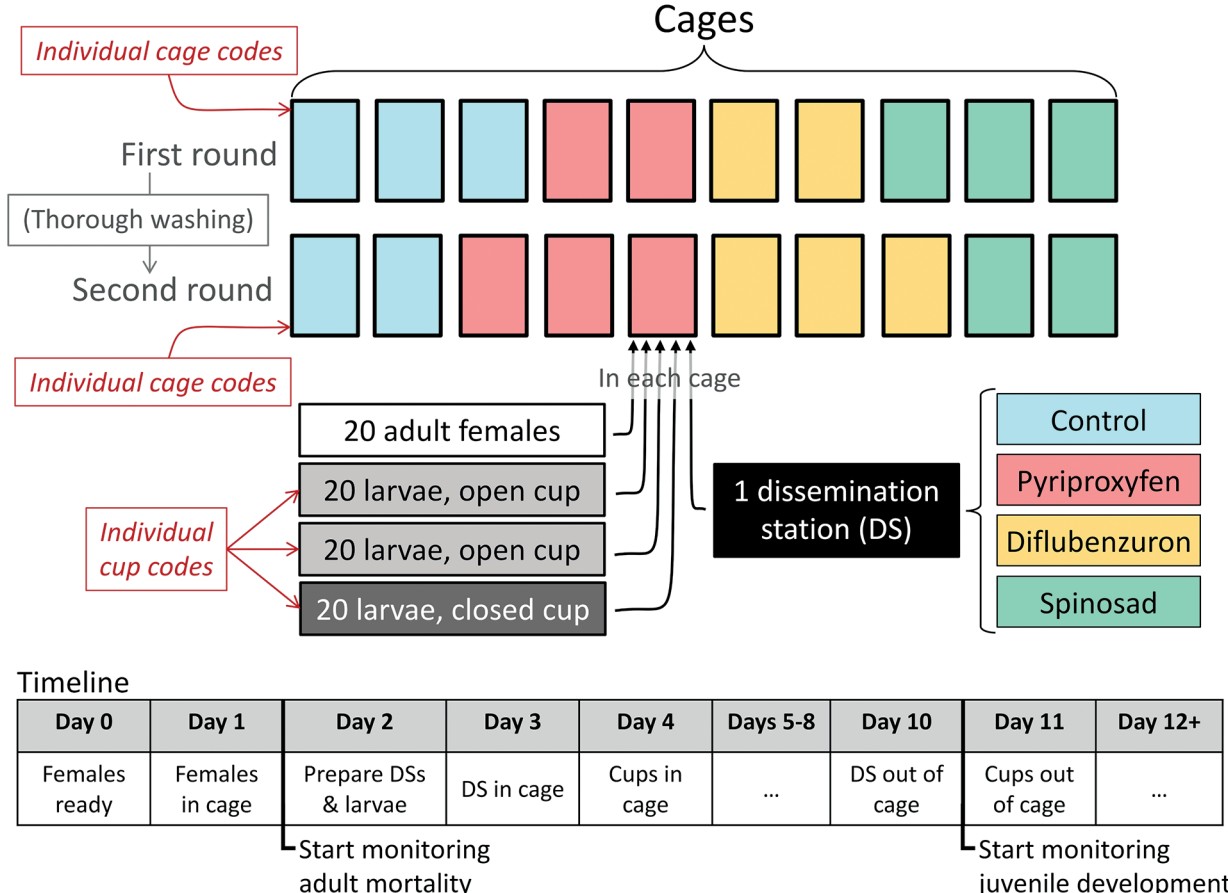

Fig. 2: design of mosquito-driven dissemination cage experiments. Ten cages were used in two rounds; each cage had one insecticide dissemination station (four treatments: pyriproxyfen, diflubenzuron, spinosad, and pumice powder), three oviposition cups (two open and one closed) with 20 *Aedes aegypti* larvae each, and 20 *Ae. aegypti* adult females. Codes were used to keep researchers evaluating mosquito outcomes (juvenile mortality, adult lifespan and death hazard) blind to treatment. The lower panel shows the experiment timeline for each of the two rounds.

avoid repetition. All these analyses were run on an intention-to-treat basis (*i.e.*, ignoring possible treatment leakage into control, closed cups; see RESULTS) in R v 4.4.2.[37] The datasets and R code needed to reproduce the results of this report are available at Figshare (https://doi.org/10.6084/m9.figshare.30265336.v1).

## RESULTS

*Diagnostic-concentration bioassays* - Juvenile mortality was 100% in all insecticide treatments, *vs.* ~9% in the control group; as expected after the mode of action of each insecticide, DFB and SPN killed mainly larvae, whereas PPF killed mainly pupae (Fig. 3). These results confirmed that the local *Ae. aegypti* population used in our experiments was susceptible to PPF, DFB, and SPN — and that our choice of pumice powder as a 'placebo' negative control was well justified.

*Mosquito-driven dissemination: cage experiments* - We monitored juvenile development in a total of 1705 *Ae. aegypti* — the 1200 larvae we placed inside oviposition cups within the experimental cages plus 505 larvae hatched from eggs laid by the mated females we released in the cages. Table I presents a summary of our obser-

vations, stratified by treatment and oviposition cup type (open or closed). Relative to the control group, there was a sharp decline in observed adult emergence from open cups in the PPF and DFB treatments, but only a modest decline in the SPN treatment. As expected, PPF induced mortality preferentially at the pupal stage, whereas DFB and (to a lesser extent) SPN induced mortality preferentially at the larval stage (Table I). Adult emergence from closed cups was also visibly reduced in PPF cages and perhaps slightly reduced in DFB cages, suggesting that some closed cups might have become contaminated. In particular, only seven of the 20 larvae (35.0%) in one closed cup set within a PPF cage completed development and emerged as adults — a value that is apparently compatible with those seen among open cups in the same treatment (32.7% on average; score 95% CI, 27.5-38.5%; range, 12.5-50.0%) (Table I, Fig. 4).

To quantify treatment effects on juvenile-mosquito development, we ran binomial generalised linear mixed models (GLMMs) with adult emergence (a binary indicator for each of the $N = 1705$ juvenile mosquitoes) as a function of treatment (four-level factor), cup (binary), and their interaction, and with random intercepts for

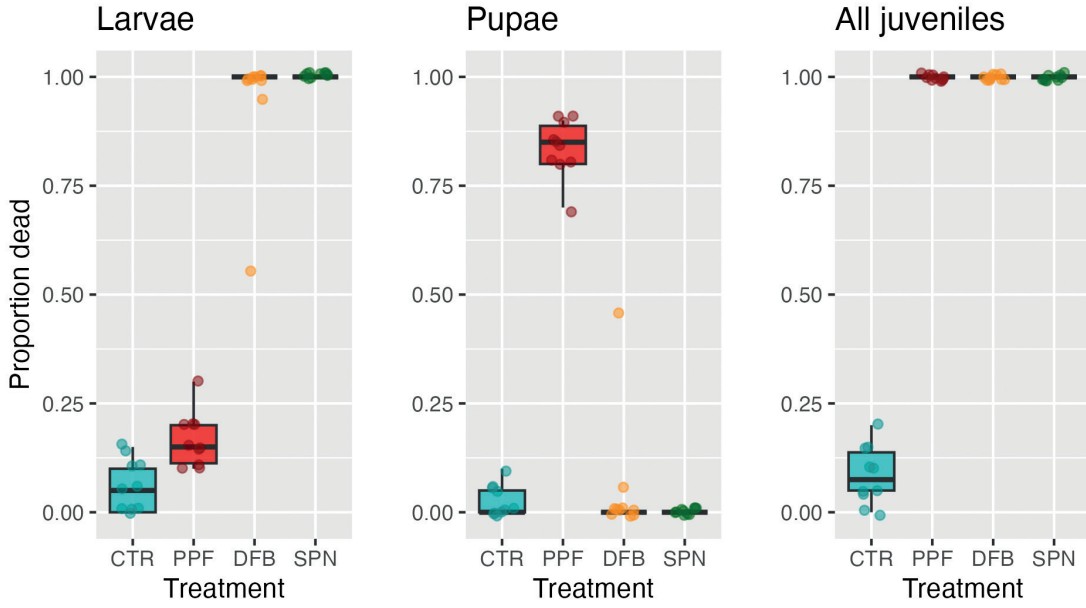

Fig. 3: diagnostic-concentration bioassay results. *Aedes aegypti* juvenile mortality (given as the proportion of larvae, pupae, and larvae + pupae that died before reaching adulthood) caused by diagnostic concentrations of pyriproxyfen (PPF), diflubenzuron (DFB), and spinosad (SPN); a negative control (CTR, pumice powder) was also tested. Circles give mortality in each of 10 replicates (20 larvae each) per treatment. Boxplots highlight median values (thick horizontal line) and inter-quartile ranges (IQR; boxes); upper whiskers extend from the upper quartile (Q3) to either Q3 + 1.5 × IQR or the maximum observed value (whichever is smaller), and lower whiskers from the lower quartile (Q1) to either Q1 - 1.5 × IQR or the minimum observed value (whichever is larger).

TABLE I

Effects of mosquito-disseminated pyriproxyfen (PPF), diflubenzuron (DFB), and spinosad (SPN) on *Aedes aegypti* juvenile (larval and pupal) mortality and adult emergence: observed results, given as number (proportion of total), from cage experiments including a negative control group (CTR)

| Treatment | Cup* | Total juveniles | Dead as larvae | Dead as pupae | Emerging as adults |
|---|---|---|---|---|---|
| CTR | Open | 362 | 42 (0.116) | 5 (0.014) | 315 (0.870) |
| | Closed | 100 | 4 (0.040) | 6 (0.060) | 90 (0.900) |
| PPF | Open | 278 | 50 (0.180) | 137 (0.493) | 91 (0.327) |
| | Closed | 100 | 19 (0.190) | 12 (0.120) | 69 (0.690) |
| DFB | Open | 407 | 171 (0.420) | 41 (0.101) | 195 (0.479) |
| | Closed | 100 | 10 (0.100) | 11 (0.110) | 79 (0.790) |
| SPN | Open | 258 | 37 (0.143) | 23 (0.089) | 198 (0.767) |
| | Closed | 100 | 7 (0.070) | 0 (0.000) | 93 (0.930) |

*oviposition cups placed inside cages; 'open' cups (10 per treatment) were open and hence available for contamination by egg-laying females, whereas 'closed' cups (five per treatment) were closed with nonwoven fabric (from disposable lab coats) and acted as 'in-cage' negative controls.

individual cage (a 20-level factor) and cup (a 60-level factor) to account for cage- and cup-level dependencies. Using BIC scores, we identified juvenile-mosquito crowding (the scaled number of juveniles developing in each cup) as an important covariate; in contrast, differences between the two experimental rounds, as well as temperature and RH effects, were not significant. The numerical output of our focal, smallest-BIC model, which includes crowding as a covariate, is presented in Table II, and model-predicted probabilities of adult-mosquito emergence from open and closed cups

across treatments and for two levels of juvenile-mosquito crowding are presented in Fig. 5. These results show that the effectiveness of mosquito-driven dissemination declined from PPF to DFB to SPN. They also suggest that there may have been some contamination of closed cups in cages with PPF, and perhaps also in cages with DFB; thus, our results likely underestimate to some extent the effectiveness of mosquito-driven dissemination for those two insecticides. There was statistical evidence of mosquito-driven dissemination of SPN, but the estimated impact on adult-mosquito emergence was small.

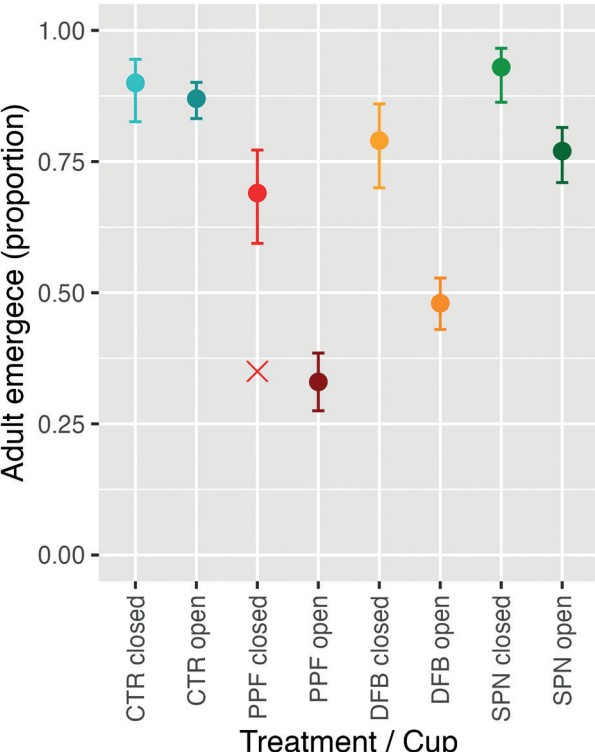

Fig. 4: observed effects of mosquito-disseminated pyriproxyfen (PPF), diflubenzuron (DFB), and spinosad (SPN) on *Aedes aegypti* adult emergence: results from cage experiments including a negative control group (CTR). For each treatment, we show the average (circle) and the score 95% confidence interval (CI; error bars) both for closed (lighter shade; 'in-cage' negative controls, five per treatment) and open oviposition cups (darker shade; 10 per treatment). The red '×' symbol highlights low adult-emergence from an 'in-cage' negative control (closed) cup that likely became contaminated with PPF; if we exclude this point from the calculations, average adult emergence from closed cups inside PPF cages rises to 0.78 (95% CI, 0.67-0.85) — almost the same as for DFB (0.79; 95%CI, 0.70-0.86).

Finally, there was an also small, negative effect (which our model adjusts for) of juvenile-mosquito crowding on juvenile survival to adulthood (Table II, Fig. 5).

We also aimed at measuring the effects of exposure to insecticide-treated DSs on the lifespan of *Ae. aegypti* adult females ($N = 400$). We first tabulated survival data and plotted Kaplan-Meier survival curves to show that, relative to exposure to sham-treated DSs (CTR), exposure to SPN-treated DSs clearly shortened female lifespan, whereas exposure to PPF- or DFB-treated DSs did not (Fig. 6). Fig. 6 also shows that mortality rates in SPN cages were particularly high early in the experiments, and especially over the seven days when DSs were still within the cages (purple band in Fig. 6). This suggests that contact with SPN-treated DS surfaces was quickly lethal for many adult females.

To get adjusted estimates of instantaneous death-hazards across treatments, we ran a series of Cox proportional-hazards mixed-effects models with treatment fixed effects, cage-level random intercepts, and varying covariate structure. BIC-score comparisons selected a model including (scaled) minimum temperature

('t_min') and maximum RH ('rh_max') as covariates; Table III shows the numerical output of this model. The clearly positive effect of SPN corresponds to an adjusted death-hazard ratio of 2.42 (95% CI, 1.17-5.01), relative to the CTR group baseline hazard; Fig. 7 shows adjusted death hazards as predicted by the model at the observed average t_min (25.6ºC) and rh_max (88.2%).

## DISCUSSION

The large-scale deployment of insecticides to control disease vectors often leads to the selection of resistant vector populations; insecticide-resistance management is, therefore, a key component of rational vector control. [20-22,38-40] As novel insecticide-based control strategies are developed, resistance management critically depends on the availability of active ingredients with diverse modes of action and favourable product profiles — *i.e.*, suitably formulated, safe, and cost-effective products. [38,39,41] One promising novel strategy for urban mosquito control is mosquito-driven dissemination of insecticides, [42] yet just one active ingredient — the juvenile-hormone analogue, PPF — has so far been shown to be effective in field trials. [13-19] Using replicate, blind, controlled cage experiments, here we tested two alternative mode-of-action products potentially suitable for mosquito-driven dissemination — a chitin-synthesis inhibitor (DFB) [26] and a biological neurotoxin composite (SPN). [27] We found that *Ae. aegypti* females can disseminate DFB and SPN to untreated larval habitats, leading to reduced adult-mosquito emergence. Exposure to SPN, however, quickly killed many adult females — an acute-toxicity effect that apparently hampered SPN dissemination. DFB thus emerges as a potentially useful alternative or complement to PPF for mosquito-driven dissemination in the context of insecticide-resistance management based on rotations or combinations. [22] While DFB and PPF act on different biological processes and have different modes of action, [43] their detoxification may involve some common biochemical pathways (*e.g.*, some cytochrome P450 enzymes); [44] hence, resistance monitoring should consider not only resistance to PPF or DFB separately, but also the possibility of PPF/DFB cross-resistance — which, to our knowledge, has so far not been identified in any *Aedes* field population.

We report a moderate reduction of adult-mosquito emergence from open oviposition cups set inside cages with one DFB-treated dissemination station (DS). At a density of ~33 larvae per 100 mL of water (the observed average in open cups), our focal adult-emergence model (Table II) predicts that just ~56.0% (95%CI, 41.9-69.1%) of *Ae. aegypti* juveniles will complete development, *vs.* a much higher 90.3% (95%CI, 83.5-94.5%) in control cages; by contrast, exposure to SPN-treated DSs reduced adult emergence more modestly (to a predicted average of 75.2%; 95%CI, 62.8-84.5%), and exposure to the much more potent PPF had the strongest effect (30.4% predicted emergence; 95%CI, 20.0-43.3% — overall consistent with the results of previous field trials) [13,14,15,16] (Fig. 5). Because we found some evidence suggesting leakage of PPF, and perhaps DFB (Figs. 4-5, Table II), into 'in-cage' negative controls (closed cups set inside test cages), our

TABLE II

Effects of mosquito-disseminated pyriproxyfen (PPF), diflubenzuron (DFB), and spinosad (SPN)
on *Aedes aegypti* adult emergence: numerical output of a generalised linear mixed model

| Term | Estimate | SE | z | p | CI lower | CI upper |
|---|---|---|---|---|---|---|
| Fixed effects | | | | | | |
| Intercept | 2.189 | 0.461 | 4.75 | - | 1.286 | 3.091 |
| PPF[a] | -1.498 | 0.580 | -2.53 | 0.0098 | -2.636 | -0.361 |
| DFB | -0.937 | 0.593 | -1.58 | 0.1138 | -2.098 | 0.224 |
| SPN | 0.369 | 0.670 | 0.55 | 0.5821 | -0.944 | 1.681 |
| Open cup | 0.138 | 0.494 | 0.28 | 0.7798 | -0.831 | 1.107 |
| PPF × open cup | -1.565 | 0.592 | -2.65 | 0.0082 | -2.725 | -0.405 |
| DFB × open cup | -1.059 | 0.594 | -1.78 | 0.0743 | -2.223 | 0.104 |
| SPN × open cup | -1.496 | 0.687 | -2.18 | 0.0295 | -2.842 | -0.149 |
| Crowding (scaled)[b] | -0.228 | 0.114 | -1.99 | 0.0465 | -0.451 | -0.004 |
| Random intercepts (SD) | | | | | | |
| Cup (*n* = 60) | 0.442 | - | - | - | 0.275 | 0.711 |
| Cage (*n* = 20) | 0.459 | - | - | - | 0.260 | 0.808 |

*a*: the clearly negative coefficient of this main term suggests that there was contamination of closed oviposition cups in cages with PPF-treated dissemination stations (see also Fig. 4); *b*: number of juveniles developing in each cup, scaled to mean 0 and SD 1 (observed mean = 28.42; observed SD = 11.44); SE: standard error; z: z-statistic; p: p-value associated with z; CI lower and CI upper: lower and upper limits of the 95% confidence interval; SD: standard deviation.

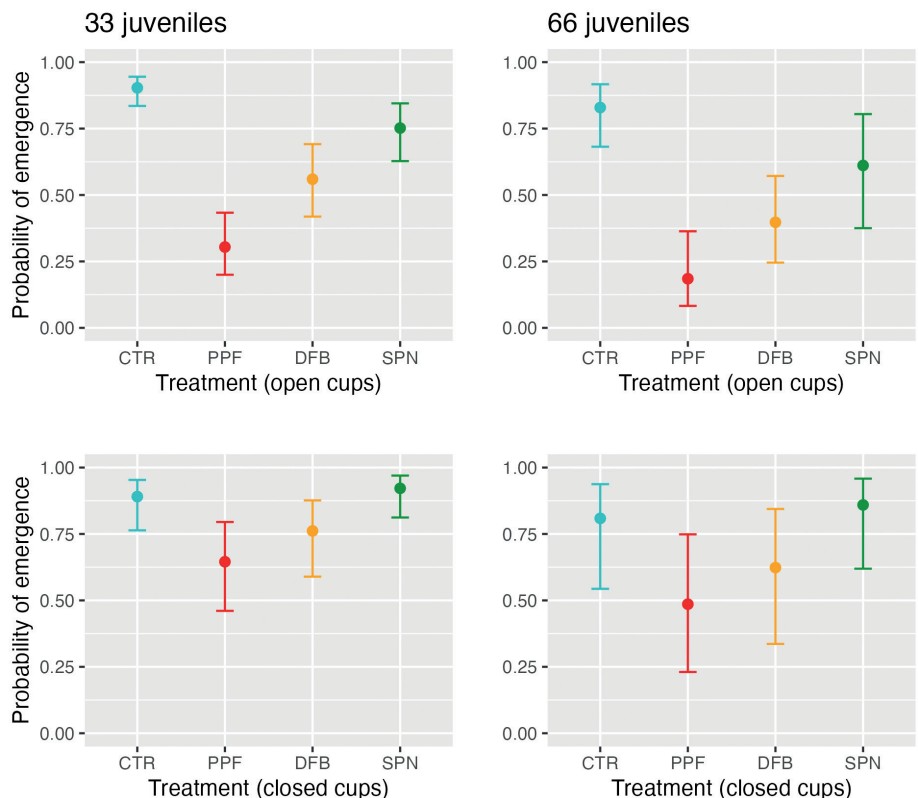

Fig. 5: effects of mosquito-disseminated pyriproxyfen (PPF), diflubenzuron (DFB), and spinosad (SPN), compared to a negative control (pumice powder; CTR), on *Aedes aegypti* adult emergence from open (upper row) and closed oviposition cups (lower row): predictions from a binomial generalised linear mixed model. Circles show predicted probabilities of adult-mosquito emergence [error bars, 95% confidence intervals (CI)]; to illustrate the small additive, negative effect of crowding on juvenile survival, we present predictions at the average observed juvenile-mosquito density in open cups during our experiments (33 juveniles per 100 mL of water) and twice that value.

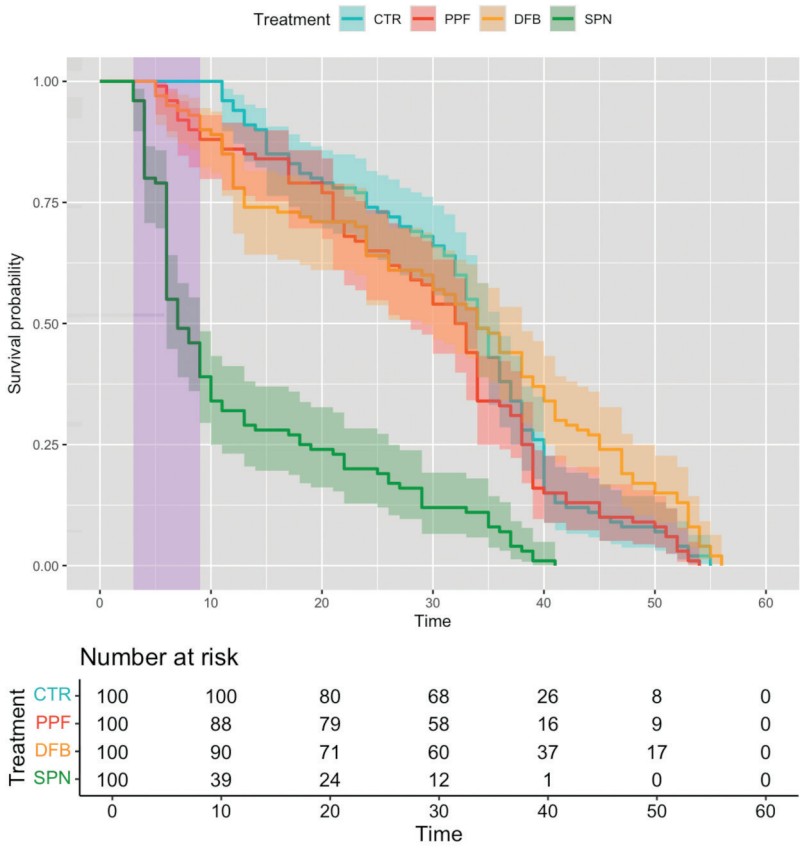

Fig. 6: effects of exposure to dissemination stations (DSs) treated with pyriproxyfen (PPF), diflubenzuron (DFB), or spinosad (SPN), compared to a negative control (pumice powder; CTR), on the lifespan of caged *Aedes aegypti* adult females: Kaplan-Meier survival curves (coloured lines) and log-log 95% confidence intervals (coloured strips around lines) over up to 60 days ('Time', x-axis) of daily observations. The purple vertical band at days 3-9 highlights the time-period during which DSs remained inside the cages. The lower table shows the numbers of females that were still alive at seven time-points in each treatment.

estimates of adult-emergence inhibition by these two compounds are likely underestimates. On the other hand, our model-based estimates adjust for the effects of larval crowding on juvenile development — which, in line with prior studies,[45,46] were negative: odds ratio = 0.797 (95%CI, 0.637-0.996) for each 1 SD (~11.4 larvae per 100 mL of water) increase in larval density (Table II, Fig. 5).

One important feature of candidate molecules for mosquito-driven dissemination is that they should not cause acute toxicity in adult mosquitoes — if adults visiting a treated DS are quickly killed, knocked-out, or otherwise rendered unable to visit further larval habitats, then dissemination can be compromised. To see if exposure to treated DSs affected adult lifespan, we measured female-mosquito survival (*N* = 400 *Ae. aegypti* females) over 60 days in our experimental cages. We expected that exposure to SPN-treated DSs would shorten adult-female lifespan to some extent, yet perhaps slowly enough not to hinder dissemination. We found, however, that SPN quickly killed many adult females, with ~61% of them (*vs.* ~0-22% in the other treatments) dying in days 3-9 of the experiment — *i.e.*, when DSs were still set inside the cages (Fig. 6). This translated into an adjusted SPN death-hazard ratio of ~2.42 (95%CI, 1.17-5.01; mixed-effects Cox model, Table III); overall female survival and death-hazards, in contrast,

were similar in pumice-treated (control) cages and in PPF- and DFB-treated cages (Fig. 6). These results suggest that while SPN might be useful as a contact adulticide in lethal ovitraps,[47,48] it is not a strong candidate for mosquito-driven dissemination — a view that is further supported by the small reduction of adult emergence from open cups in SPN-treated cages (Fig. 5).

Our study has several limitations. First, while cage-experiment results provide crucial guidance and justification for field testing, their external validity needs to be established in real-world settings. Along the same lines, we did our tests on *Ae. aegypti* mosquitoes from Manaus because that is where we plan to run small-scale field trials, but whether and to what extent the results will replicate in mosquitoes with different genetic backgrounds remains to be seen. We also note that we did not experimentally define diagnostic concentrations for each test compound; instead, we retrieved those values from the literature.[29,30,31] This was because, with that part of the study, we just aimed at making sure that (i) our mosquitoes were not resistant to the test active ingredients and (ii) those ingredients were indeed active and working as expected. Finally, we did not have the chemical-analytic means to measure active-ingredient concentrations in the water of oviposition cups set inside experimental cages; instead, we directly measured the

TABLE III

Effects of exposure to dissemination stations treated with pyriproxyfen (PPF), diflubenzuron (DFB), or spinosad (SPN) on the lifespan of caged *Aedes aegypti* adult females: numerical output of a Cox proportional-hazards mixed-effects model

| Term | Estimate | SE | *z* | p | CI lower | CI upper |
|---|---|---|---|---|---|---|
| Fixed effects | | | | | | |
| PPF | 0.105 | 0.361 | 0.29 | 0.7733 | -0.602 | 0.811 |
| DFB | -0.447 | 0.360 | -1.24 | 0.2149 | -1.152 | 0.259 |
| SPN | 0.882 | 0.372 | 2.37 | 0.0177 | 0.153 | 1.611 |
| t_min (ºC, scaled) | 1.504 | 0.112 | 13.39 | < 0.0001 | 1.284 | 1.724 |
| rh_max (%, scaled) | 1.184 | 0.118 | 10.02 | < 0.0001 | 0.952 | 1.415 |
| Random intercepts (SD)* | | | | | | |
| Cage (*n* = 20) | 0.519 | - | - | - | 0.463 | 0.778 |

*\*coxme* 2.2-22 does not provide CIs for random-effect estimates; the CI limits given here were computed using non-parametric bootstrapping (1000 replicates). SE: standard error; *z*: *z*-statistic; p: p-value associated with *z*; CI lower and CI upper: lower and upper limits of the 95% confidence interval; t_min, minimum daily temperature; rh_max, maximum daily relative humidity; SD: standard deviation.

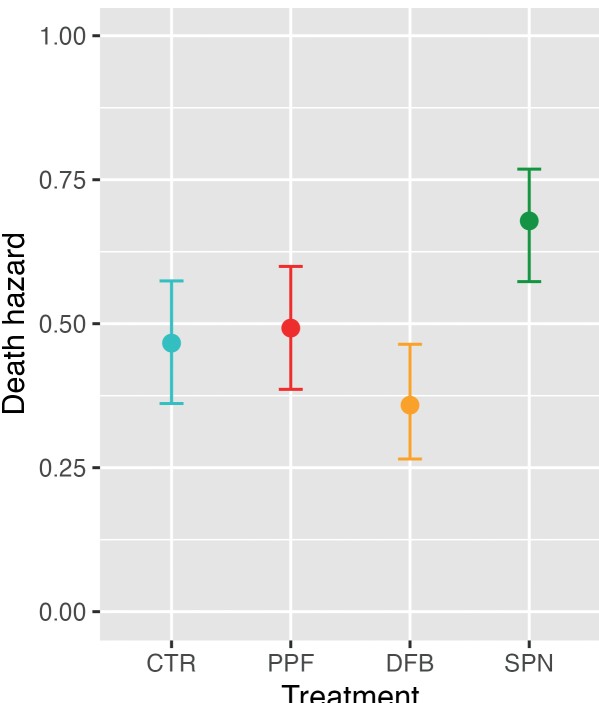

Fig. 7: effects of exposure to dissemination stations treated with pyriproxyfen (PPF), diflubenzuron (DFB), or spinosad (SPN), compared to a negative control (pumice powder; CTR), on the death hazard of caged *Aedes aegypti* adult females: predictions from a Cox mixed-effects proportional-hazards model. Predictions are given on the probability scale; they correspond to expectations at average values of minimum temperature and maximum relative humidity, and take into account the non-independence of observations made in each of 20 experimental replicates (five per treatment).

entomological endpoints of interest (juvenile mortality and adult emergence) and assumed that any changes, relative to control cages and cups, were due to mosquito-driven dissemination. Determination of active-ingredient concentrations would also have helped us firmly establish whether there was leakage of PPF and DFB into closed cups inside treated cages — and this, in turn, would have allowed us to measure the degree to which our results underestimate adult-emergence inhibition by those two compounds.

Taken together, in sum, our findings suggest that the chitin-synthesis inhibitor, DFB, but not the biological neurotoxin composite, SPN, may be useful for *Aedes* control in the context of mosquito-driven dissemination. These encouraging experimental results, together with prior theoretical work,[49] indicate that small-scale field trials should proceed with the aim of testing whether deployment of DFB-treated DSs effectively leads to mosquito-driven dissemination in real-world settings — and, if so, whether such dissemination yields suitable levels of larval-habitat coverage and leads to measurable reductions of adult-mosquito emergence and mosquito population densities. Other promising chitin-synthesis inhibitors, such as lufenuron[50] and novaluron[51] should also enter the mosquito-driven insecticide dissemination testing pipeline — from laboratory bioassays and cage experiments such as those described here to, if the results are encouraging, field trials.

## ACKNOWLEDGEMENTS

To the Fundação Nacional de Saúde (FUNASA) in Manaus for permission to use their facilities.

## AUTHORS' CONTRIBUTION

SLBL - funding acquisition, project administration, writing - review and editing; SLBL and FA-F – conceptualisation and methodology; FA-F - formal analysis, writing - original draft and visualisation; SLBL, FA-F, JJC-C and CTC - methodology, investigation and supervision; GBAA, ASG, FASF and JJC-C - methodology, investigation, data acquisition and management, formal analysis, writing - review and editing. All authors read and critically revised the manuscript for intellectual content, and all approved the final version. The funders had no role in study design, data collection and analysis, decision to publish, or preparation of the manuscript. The authors declare that they have no competing interests.

## DATA AVAILABILITY

The datasets and R code needed to reproduce the results of this report are available at Figshare (https://doi.org/10.6084/m9.figshare.30265336.v1).

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

# OPEN PEER REVIEW

Memórias do IOC thanks the anonymous reviewers for their contribution to the peer review of this work.

## FIRST REVIEW ROUND

REVIEWERS' COMMENTS

**REVIEWER #1**

Dear Editor,

This study set out to evaluate alternative larvicides for mosquito-driven dissemination, given the increasing risk of resistance to pyriproxyfen (PPF), a juvenile hormone analogue. Through simulated, blinded, and controlled cage experiments, the authors assessed the potential for Aedes aegypti females to disseminate diflubenzuron (DFB) and Spinosad (SPN), in comparison to the gold standard, PPF. The results showed that both DFB and, to a lesser extent, SPN significantly reduced adult mosquito emergence in untreated containers, though less effectively than PPF. Notably, exposure to SPN rapidly reduced adult female survival, likely limiting its potential for dissemination. The authors conclude that diflubenzuron could be a promising alternative to PPF and merits further field testing.

The manuscript is well written and addresses a timely and relevant topic, particularly given the limited number of alternative active ingredients suitable for mosquito autodissemination - a strategy the WHO considers a promising complementary tool for reducing vector populations and arbovirus transmission. The statistical analyses appear robust. However, there are several concerns that should be addressed prior to publication.

1. Rationale for insecticide selection

A core principle of the autodissemination strategy is the use of slow-acting compounds that allow female mosquitoes to survive long enough to transfer the insecticide to larval habitats. Additionally, the target mosquito population should be susceptible to the insecticide deployed at the dissemination station to minimize resistance selection and ensure the long-term efficacy of the strategy.

These considerations raise questions about the suitability of the two alternative products tested in this study:

• Spinosad (SPN) is a biopesticide with strong adulticidal activity, as confirmed by the authors and previously demonstrated in multiple studies. Given its fast-acting nature, what was the rationale for selecting SPN for a dissemination strategy, knowing that it would likely kill females before they could transfer the product to breeding sites?

• Diflubenzuron (DFB), like PPF, is an insect growth regulator (IGR) that inhibits chitin synthesis. Moderate levels of resistance to both PPF and DFB have been documented in Ae. aegypti populations, including in Brazil (Campos et al. 2020, 2023; Araújo et al. 2019). P450 monooxygenases have been implicated in PPF detoxification, and cross-resistance due to elevated P450 activity is plausible (Yunta et al. 2016). Given the similar mode of action and potential for cross-resistance, what is the justification for selecting DFB as a viable alternative for resistance management?

Would it not be more appropriate to test new compounds with unrelated modes of action to PPF, such as slow-acting pyrroles (e.g., chlorfenapyr), oxadiazines, or some neonicotinoids (e.g., imidacloprid), possibly in combination, to enhance efficacy and mitigate resistance? his This a key point, especially since the authors justified the rationale of the study within the framework of integrated vector management.

2. Resistance diagnostics

The section on diagnostic concentrations (DC) requires clarification. According to WHO guidelines (WHO, 2022), DCs should be established using a susceptible laboratory reference strain (e.g., Rockefeller, New Orleans, Bora Bora). If the same field-collected population was used both to define DCs and to assess resistance levels, the results may be unreliable or misleading. This section should be revised accordingly or removed if appropriate data are lacking.

Minor comments and clarifications:

Introduction

• Line 96: Consider rephrasing as "Our study showed that diflubenzuron..."

• Line 90: WHOPES has been replaced by WHO PQT since 2017. Please update the terminology and provide a link to the list of WHO-prequalified vector control products.

Materials and Methods

• Line 100: What was the age of the adult females used in the bioassays? Why not use a standard laboratory colony (e.g., Rockefeller) to minimize variability due to genetic background?

Mosquito-driven dissemination: cage experiments

The autodissemination station is insufficiently described. The use of plastic cups is mentioned, but the design should be detailed in the main text. Reference should be made to Figures 1 and 2 for clarity.

Results: diagnostic concentration bioassays

• If no susceptible reference strain was used to establish DCs, the methodology and conclusions on susceptibility may be invalid. Please clarify this point and revise the section if necessary.

Results

• Line 189: "Sharp decline in observed adult emergence" – please quantify the reduction and specify if it was statistically significant.

• Line 209: How do the authors explain the relatively high mortality in control cups (13%)? Could cross-contamination have occurred in the controls, similar to what has been reported for PPF?

• Despite a relatively large sample size (n = 407 over two rounds), DFB's impact on emergence was not statistically significant (47% emergence vs. 87% in controls). Could there be any bias or lack of power in the statistical model? Have you test other models?

Discussion

• L249: The authors indicate a significant reduction in precited emergence with DFB, whereas Table 2 suggests the reduction is not statistically significant (Estimate = –0.937, P = 0.1138). Please clarify this discrepancy.

• The rise of Aedes resistance to insect growth regulators (IGRs) such as PPF and DFB in Brazil should be discussed, as it may threaten the future success of the autodissemination strategy based on DFB.

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

**REVIEWER #2**

Insecticide dissemination stations, implemented relatively recently to control Aedes aegypti in Brazil, are based on the characteristic "jumping posture" of this species' females. The rationale behind this method is to ensure the spread of larvicides across different potential mosquito breeding sites by the adult females themselves.

The manuscript reports on "simulated" trials in large cages on the feasibility of using two insecticides not previously tested using this methodology with the dissemination stations: diflubenzuron, a chitin synthesis inhibitor, and spinosad, a neurotoxic insecticide that modulates acetylcholine receptors. The trials are well designed, executed, and analyzed. The team has extensive experience in developing and implementing this methodology and even includes professionals responsible for establishing "dissemination stations" for Aedes aegypti control in Brazil.

Furthermore, validation for use of alternative larvicides to pyriproxyfen (a juvenile hormone analogue) in dissemination stations is welcome, as it helps to identify A. aegypti chemical control options. It is worth noting that one of the Ministry of Health (MoH) guidelines is to encourage the rotation of products with different mechanisms of action, in order to prevent, or to delay, the spread of vector resistance, thus preserving the products used in the chemical control of Aedes aegypti.

However, I understand that the main shortcoming of the work is the lack of a more detailed technical justification and argument for the inclusion of spinosad in the tests. Diflubenzuron and pyriproxyfen interfere with physiological, or endocrinological, processes specific to the immature stages. In contrast, spinosad has the potential to also act on adults; moreover, unlike the other two products, because it is a neurotoxic agent, its action is rapid, potentially leading to the death of the adult female, precisely the specimen that performs the "dissemination work." This conclusion, in fact, was confirmed by the results of the manuscript. Exploring the mode of action of each insecticide in the Introduction, anticipating the expected outcomes of the trials, could be a way to add robustness to the text. Furthermore, detailing, in the methodology and/or results, the time employed to monitor the larvae treated with each of the different products helps to highlight the differences in their modes of action.

Below, observations in the order they appear in the text.

Abstract, "Alternative larvicide/pupicide molecules suitable for mosquito-driven dissemination, but with distinct modes of action, are therefore needed."- It is worth mentioning (in this section or in the Introduction) that this initiative is in line with the recommendations of the MoH, which advocate rotation of products with different modes of action, as a way of ensuring greater longevity of the available insecticides, preserving them.

Introduction

I suggest commenting in this section that dissemination stations are successful in controlling Aedes aegypti because they exploit two unique characteristics of adult females of this species: (1) they spread their eggs across many breeding sites, and (2) they prefer artificial containers, either inside or outside home, used by humans.

- on the requirements for insecticides to work in dissemination stations:

- lines 82-83, "(iii) are formulated as powder particles suitable for mosquito-driven dissemination" - Is there a chance that a liquid formulation would work? Couldn't such a formulation also be impregnated into the female mosquito?

- lines 83-84, "(iv) have mechanisms of action that differ from that of the insecticide targeted for management" - I didn't understand. Wouldn't this strategy be part of management? Or is the text referring to 'adulticides targeted for management'?

Materials and methods

Item 'Diagnostic-concentration bioassays' (starts on line 120): as already mentioned, I suggest providing more details about the monitoring time, or the average mortality time, of larvae exposed to each product.

– Line 155, "Adult mosquitoes remained inside the cages until death or for up to 60 days." - inform that (if) the females received any source of sugar as food during this period.

Results

- Item 'Diagnostic-concentration bioassays', lines 178-183 - It would be interesting to include information on the time dynamics, or mean time, of mortality with each product tested, since their modes of action are distinct. DFB is supposed to impact each molt, while the expected action of SPN, a neurotoxic agent, is quick. Even in the standard assays recommended for these products, there are differences in follow-up time (generally 24 hours for neurotoxic agents and several days for IGR ones).

Discussion

Lines 247-248, "DFB thus emerges as a potentially useful alternative or complement to PPF for mosquito-driven dissemination" - insecticide rotation in order to preserve them.

Line 257, reference to "Because we found some evidence suggesting leakage of PPF" - Judging by Figure 4, leakage appears to have occurred for all three products tested. To what factor do the authors attribute this effect? Some technical problem with any undesirable 'opening' of the 'closed cups' that allowed adult females to enter? Or to the spread of the products in the cage environment by females during flight?

Lines 285-286, "Cage tests should proceed in parallel to investigate the potential usefulness of combining DFB with PPF" – It's worth considering (and discussing) that this field initiative involving the simultaneous use of two IGRs has the potential to accelerate the development and spread of resistance to both compounds. It's a way to prioritize chemical control. While it may have a significant effect in the short term, it risks compromising more sustainable strategies, in which chemical vector control should be used as a complement to vector control.

A curiosity: it would be pertinent to include, in the Discussion, information on the average effectiveness of MD-PPF (in terms of adult emergence inhibition) currently in field use. Additionally, it is worth discussing, based on (a) the performance, in the "simulated field" trials, of each of the insecticides presented in this Manuscript, and (b) the effectiveness of MD-PPF already in field use, whether it is possible to estimate, or envision, the effect of implementing MD with other products (mainly DFB) in routine control.

Minor comments

Table I, legend, "(from disposable lab coats)" – This is already included in the methodology. There's no need to repeat it

Discussion, line 258, "DFB (Fig. 2, Table II)" - or Figure 4?

**REVIEWER #3**

a) Adequacy of the abstract; b) Originality and importance of the contribution for the development of the field of study. And relevance of: c) Methodology, results and discussion; d) References; e) Figures and tables.

<div align="right">AUTHORS' RESPONSE TO THE REVIEWERS</div>

Belo Horizonte, 20/08/2025

Dear Dr Martins,

we first want to thank you and the Reviewers for the detailed, overall positive assessment of our work, for the constructive criticism, and for the useful comments and suggestions.

In particular, we appreciate the opportunity to clarify the rationale for testing spinosad – a point that, we agree, was not well explained in the first version of the manuscript. In short (see also below and R1 version), we wanted to test if, and to what extent, spinosad shortened adult lifespan in the specific context of autodissemination. The idea was that, if spinosad killed females only slowly (too slowly to significantly hinder dissemination), then it could hold promise for autodissemination – similar to (but simpler than) the use of slow-killing entomopathogenic fungi in combination with PPF in autodissemination stations.

We also take the opportunity to clarify that we did not experimentally define bioassay Diagnostic Concentrations (DCs). This is because, with that part of the study, we just aimed at making sure that (1) our mosquitoes were not resistant to the active ingredients we used and (2) those ingredients were indeed active and working as expected. For that, we simply retrieved DCs from the literature – as indicated in l. 128-129: "...based on published results, we used 0.02 parts per million (ppm) for PPF, 1.5 ppm for DFB, and 1 ppm for SPN[29-31]...", where the refs. are:

29. Andrighetti MTM, Cerone F, Rigueti M, Galvani KC, Macoris MLG. Effect of pyriproxyfen in Aedes aegypti populations with different levels of susceptibility to the organophosphate temephos. Dengue Bull. 2008; 32: 186-98.

30. Montaño-Reyes A, Llanderal-Cazares C, Valdez-Carrasco J, Miranda-Perkins K, Sanchez-Arroyo H. Susceptibility and alterations by diflubenzuron in larvae of Aedes aegypti. Arch Insect Biochem Physiol. 2019; 102: e21604.

31. Aldridge RL, Alto BW, Connelly CR, Okech B, Siegfried B, Linthicum KJ. Lethal and sublethal concentrations of formulated larvicides against susceptible Aedes aegypti. J Am Mosq Control Assoc. 2022; 38: 250-60.

We have added a full paragraph (l. 294-311) explicitly covering the main limitations of the study, including the point on DCs. Finally, we have addressed each of the points raised by the Reviewers. Below you can find, in red font, our point-by-point responses – with line numbers for any changes made to the R1 manuscript, in which all such changes are highlighted in yellow.

We hope that you will find that this revised (and clearly improved, we believe) version of our manuscript meets the standards of Memórias, and look forward to hearing from you.

Sincerely,

Fernando Abad-Franch (on behalf of all co-authors):

## REVIEWER COMMENTS

**REVIEWER #1**

Dear Editor,

This study set out to evaluate alternative larvicides for mosquito-driven dissemination, given the increasing risk of resistance to pyriproxyfen (PPF), a juvenile hormone analogue. Through simulated, blinded, and controlled cage experiments, the authors assessed the potential for Aedes aegypti females to disseminate diflubenzuron (DFB) and Spinosad (SPN), in comparison to the gold standard, PPF. The results showed that both DFB and, to a lesser extent, SPN significantly reduced adult mosquito emergence in untreated containers, though less effectively than PPF. Notably, exposure to SPN rapidly reduced adult female survival, likely limiting its potential for dissemination. The authors conclude that diflubenzuron could be a promising alternative to PPF and merits further field testing.

The manuscript is well written and addresses a timely and relevant topic, particularly given the limited number of alternative active ingredients suitable for mosquito autodissemination – a strategy the WHO considers a promising complementary tool for reducing vector populations and arbovirus transmission. The statistical analyses appear robust. However, there are several concerns that should be addressed prior to publication.

We thank the Reviewer for their clear summary of our work and positive comments.

1. Rationale for insecticide selection A core principle of the autodissemination strategy is the use of slow-acting compounds that allow female mosquitoes to survive long enough to transfer the insecticide to larval habitats. Additionally, the target mosquito population should be susceptible to the insecticide deployed at the dissemination station to minimize resistance selection and ensure the long-term efficacy of the strategy.

These considerations raise questions about the suitability of the two alternative products tested in this study:

• Spinosad (SPN) is a biopesticide with strong adulticidal activity, as confirmed by the authors and previously demonstrated in multiple studies. Given its fast-acting nature, what was the rationale for selecting SPN for a dissemination strategy, knowing that it would likely kill females before they could transfer the product to breeding sites?

This is an important point that we agree we did not explain clearly in our original submission. We first wanted to test if, and to what extent, SPN was capable of shortening adult lifespan in the specific context of autodissemination – i.e., after 'non-forced' contact with dust-treated surfaces, thus mimicking 'real-life' conditions in which mosquitoes land and possibly walk on the walls of a potential breeding site. The idea was that, if spinosad killed females only slowly 3 (too slowly to significantly hinder dissemination), then it could hold promise for autodissemination – similar to (but simpler than) the use of slow-killing entomopathogenic fungi in combination with PPF in autodissemination stations (cf. ref. 19). We found out that this is not the case – and, as a 'side effect', demonstrated that spinosad might be interesting for use in lethal ovitraps (l. 290-291). We have tried to clarify the rationale for our choice of testing spinosad (l. 98-100, 224-225, and 283-284; see also l. 230-231, 254-255, 290-293).

• Diflubenzuron (DFB), like PPF, is an insect growth regulator (IGR) that inhibits chitin synthesis. Moderate levels of resistance to both PPF and DFB have been documented in Ae. aegypti populations, including in Brazil (Campos et al. 2020, 2023; Araújo et al. 2019). P450 monooxygenases have been implicated in PPF detoxification, and cross-resistance due to elevated P450 activity is plausible (Yunta et al. 2016). Given the similar mode of action and potential for cross-resistance, what is the justification for selecting DFB as a viable alternative for resistance management?

While we are aware that detoxification of DFB and PPF may involve some common biochemical pathways, we:

(a) Note that DFB and PPF have distinct modes of action (as defined by the Insecticide Resistance Action Committee – IRAC; see https://irac-online.org/mode-of-action/); and

(b) Know of no evidence of DFB/PPF cross-resistance in Aedes.

All in all, even if some level of cross-resistance happens to emerge (a possibility we now stress), having two molecules with distinct modes of action (as defined by IRAC) available for use is still a clear advantage. We have clarified these points in l. 257-262 (and added the new refs. 43,44).

Would it not be more appropriate to test new compounds with unrelated modes of action to PPF, such as slow-acting pyrroles (e.g., chlorfenapyr), oxadiazines, or some neonicotinoids (e.g., imidacloprid), possibly in combination, to enhance efficacy and mitigate resistance? This is a key point, especially since the authors justified the rationale of the study within the framework of integrated vector management.

We are indeed considering and testing further compounds – with an eye on those that are approved for use in drinking water, act at low doses, and are formulated as (or can be micronized to) fine powder suitable for dissemination by mosquitoes (l. 80-89 and 245-246). We mention a few of them (l. 319-322), but this is currently work in progress.

2. Resistance diagnostics

The section on diagnostic concentrations (DC) requires clarification. According to WHO guidelines (WHO, 2022), DCs should be established using a susceptible laboratory reference strain (e.g., Rockefeller, New Orleans, Bora Bora). If the same field-collected population was used both to define DCs and to assess resistance levels, the results may be unreliable or misleading. This section should be revised accordingly or removed if appropriate data are lacking.

 Please note that the only aim of this part of the study was to confirm that the products available to us indeed killed mosquitoes from our colony. We did not aim at estimating, and did not estimate, our bioassay DCs – we simply retrieved them from the literature (refs. 29-31), as stated explicitly in l. 127-129.

Minor comments and clarifications:

Introduction

● Line 96: Consider rephrasing as "Our study showed that diflubenzuron…"

We have rephrased the end of the Introduction (please see l. 98-102).

● Line 90: WHOPES has been replaced by WHO PQT since 2017. Please update the terminology and provide a link to the list of WHO-prequalified vector control products.

Many thanks for this suggestion; we have now updated the terminology and provide specific refs. and links (refs. 24-27).

Materials and Methods

● Line 100: What was the age of the adult females used in the bioassays? Why not use a standard laboratory colony (e.g., Rockefeller) to minimize variability due to genetic background?

Females were 7-10 days old (see l. 151-152). We used field-derived females because we wanted to know if our test compounds worked with local populations from the sites where we conduct small-scale field tests (e.g., ref. 15); we now clarify this (l. 296-299).

Mosquito-driven dissemination: cage experiments

● The autodissemination station is insufficiently described. The use of plastic cups is mentioned, but the design should be detailed in the main text. Reference should be made to Figures 1 and 2 for clarity.

We now make explicit reference to Figs. 1 and 2 and refer readers to refs. 13 and 15 for a more detailed description of DSs (l. 150-155).

Results: diagnostic concentration bioassays

● If no susceptible reference strain was used to establish DCs, the methodology and conclusions on susceptibility may be invalid. Please clarify this point and revise the section if necessary.

Please see our response on this point above. To reiterate, the only aim of this part of the study was to confirm that the products available to us indeed killed mosquitoes from our colony. We did not aim at estimating, and did not estimate, our bioassay DCs – we simply retrieved them from the literature (refs. 29-31), as stated explicitly in l. 127-129.

Results

• Line 189: "Sharp decline in observed adult emergence" – please quantify the reduction and specify if it was statistically significant.

Please note that the numerical results are given in Table I (observed values) and Table II (numerical output from the top-performing GLMM, including estimates, standard errors, zstatistics, p-values, and 95% CI limits). Fig. 4 shows observed proportions (with score 95% CIs), and Fig. 5 shows predictions (on the probability scale) from the top-performing GLMM, with 95% CIs. We can include a summary of those results in the text, but this might seem redundant.

• Line 209: How do the authors explain the relatively high mortality in control cups (13%)? Could cross-contamination have occurred in the controls, similar to what has been reported for PPF?

We think it unlikely that any of the open cups in control cages might have become contaminated – none of them had the sharp increase in juvenile mortality we'd expect if that were the case. Mortality in closed cups within control cages was ~10%, and the small increase (to ~13%) in open cups can be parsimoniously attributed to higher crowding in open cups – where females laid eggs which developed into larvae. In particular, the highest mortality in an open cup set within a control cage was 33.3%, and this was the open cup with the largest number of larvae (57, of which 18 died as larvae and 1 as a pupa). Excluding this cup, the average mortality was identical (10%) to that seen in closed cups within control cages.

• Despite a relatively large sample size (n = 407 over two rounds), DFB's impact on emergence was not statistically significant (47% emergence vs. 87% in controls). Could there be any bias or lack of power in the statistical model?

Have you test other models?

We follow leading statisticians (see, e.g., among many recent relevant papers, https://www.nature.com/articles/d41586-019-00857-9; https://peerj.com/articles/3544/; https://www.nature.com/articles/s41562-017-0224-0 – and

some older ones, e.g., Cohen's classic https://users.cla.umn.edu/~nwaller/prelim/cohenearthround.pdf) in avoiding the use of 'statistical significance' to learn from our data. We believe that a reduction from 90% to less than 56% (the GLMM estimates most compatible with our data, at mean juvenile crowding) is biologically significant – and perhaps also significant in practice, although this remains to be tested in real-world scenarios (l. 45-46 and 312-319). Please see also our next response.

Discussion

• L249: The authors indicate a significant reduction in precited emergence with DFB, whereas Table 2 suggests the reduction is not statistically significant (Estimate = –0.937, P = 0.1138). Please clarify this discrepancy.

We meant significant in the ordinary (not 'statistical') sense of the word. This was because we believe that reducing emergence from ~90% (control cages) to ~56% (with an upper 95%CI limit of 69%) is 'significant'. To avoid confusion, however, we have removed "significant" from that sentence (l. 263).

We note, in any case, that the coefficient and p-value the Reviewer highlights are for the effect of DFB in closed cups – the main effect in a model with an interaction between cup type (closed, open) and treatment (CTR, DFB, SPN, PPF). The coefficient for the interaction (-1.059, SE 0.594 is associated with a much smaller p-value of 0.074). The 'effects' of both coefficients (main effect, interaction), together with that of the 'cup type' and 'crowding' main effects (and of course the intercept), yield the predictions we highlight – which we do precisely because interpreting predictions on a natural scale (here, %) is usually much more intuitive than interpreting (link-scale) model coefficients, especially in the presence of interactions.

(Note also that the interaction coefficient estimates the net effect of DFB in open cups; exponentiating the point estimate and 95%CI limits, our best estimate is that DFB caused a reduction of ~65.3% in the odds of emergence, with the CI spanning from a ~89.2% reduction to an 11.0% increase, relative to the difference between closed and open cups in the CTR treatment – where the model estimates a ~14.8% increase in the odds of emergence in open compared to closed cups [where the model predicts emergence at $\exp(2.189)/(1+(\exp(2.189)))$ 0.89926, or ~89.9%, when juvenile crowding is ~28 per 100 mL of water].)

• The rise of Aedes resistance to insect growth regulators (IGRs) such as PPF and DFB in Brazil should be discussed, as it may threaten the future success of the autodissemination strategy based on DFB.

As mentioned above, we now discuss both resistance to individual compounds and the possibility of cross-resistance (see l. 257-262).

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

## REVIEWER #2

Insecticide dissemination stations, implemented relatively recently to control Aedes aegypti in Brazil, are based on the characteristic "jumping posture" of this species' females. The rationale behind this method is to ensure the spread of larvicides across different potential mosquito breeding sites by the adult females themselves.

The manuscript reports on "simulated" trials in large cages on the feasibility of using two insecticides not previously tested using this methodology with the dissemination stations: diflubenzuron, a chitin synthesis inhibitor, and spinosad, a neurotoxic insecticide that modulates acetylcholine receptors. The trials are well designed, executed, and analyzed. The team has extensive experience in developing and implementing this methodology and even includes professionals responsible for establishing "dissemination stations" for Aedes aegypti control in Brazil.

Furthermore, validation for use of alternative larvicides to pyriproxyfen (a juvenile hormone analogue) in dissemination stations is welcome, as it helps to identify A. aegypti chemical control options. It is worth noting that one of the Ministry of Health (MoH) guidelines is to encourage the rotation of products with different mechanisms of action, in order to prevent, or to delay, the spread of vector resistance, thus preserving the products used in the chemical control of Aedes aegypti.

We also thank Reviewer #2 for their positive remarks on our report.

However, I understand that the main shortcoming of the work is the lack of a more detailed technical justification and argument for the inclusion of spinosad in the tests. Diflubenzuron and pyriproxyfen interfere with physiological, or endocrinological, processes specific to the immature stages. In contrast, spinosad has the potential to also act on adults; moreover, unlike the other two products, because it is a neurotoxic agent, its action is rapid, potentially leading to the death of the adult female, precisely the specimen that performs the "dissemination work." This conclusion, in fact, was confirmed by the results of the manuscript. Exploring the mode of action of each insecticide in the Introduction, anticipating the expected outcomes of the trials, could be a way to add robustness to the text. Furthermore, detailing, in the methodology and/or results, the time employed to monitor the larvae treated with each of the different products helps to highlight the differences in their modes of action.

This is an important point that was also raised by Reviewer #1 and that, as discussed in our response to their comments, we agree we did not explain clearly in our original submission. Our response to Reviewer #1 reads: "We first wanted to test if, and to what extent, SPN was capable of shortening adult lifespan in the specific context of autodissemination – i.e., after 'non-forced' contact with dust-treated surfaces, thus mimicking 'real-life' conditions in which mosquitoes land and possibly walk on the walls of a potential breeding site. The idea was that, if spinosad killed females only slowly (too slowly to significantly hinder dissemination), then it could hold promise for auto-dissemination – similar to (but simpler than) the use of slow-killing 8 entomopathogenic fungi in combination with PPF in autodissemination stations (cf. ref. 19). We found out that this is not the case – and, as a 'side effect', demonstrated that spinosad might be interesting for use in lethal ovitraps (l. 290-291). We have tried to clarify the rationale for our choice of testing spinosad (l. 98-100, 224-225, and 283-284; see also l. 230-231, 254-255, 290- 293)."

We followed all larvae/pupae until emergence or death (l. 140-143 and 164), but did not record the time to event in every individual juvenile. This was only done for studying the lifespan of adult females.

Below, observations in the order they appear in the text.

Abstract, "Alternative larvicide/pupicide molecules suitable for mosquito-driven dissemination, but with distinct modes of action, are therefore needed."- It is worth mentioning (in this section or in the Introduction) that this initiative is in line with the recommendations of the MoH, which advocate rotation of products with different modes of action, as a way of ensuring greater longevity of the available insecticides, preserving them.

The recommendations by the Brazilian MoH are only available in Portuguese (the most recent, at https://www.gov.br/saude/pt-br/centrais-de-conteudo/publicacoes/estudos-e-notasinformativas/2024/nota-informativa-no-29-2024-cgarb-dedt-svsa-ms). We think that refs. 20- 23, and especially ref. 22 (https://www.nature.com/articles/s41598-021-03367-9.pdf) provide a much more general and widely accessible overview on insecticide resistance and its management – including the importance of "the availability of active ingredients with diverse modes of action and that can be deployed safely and conveniently at scale" (l. 79-80).

Introduction

I suggest commenting in this section that dissemination stations are successful in controlling Aedes aegypti because they exploit two unique characteristics of adult females of this species: (1) they spread their eggs across many breeding sites, and (2) they prefer artificial containers, either inside or outside home, used by humans.

Thank you for this suggestion; we now mention those characteristics in l. 71-72.

- on the requirements for insecticides to work in dissemination stations:

- lines 82-83, "(iii) are formulated as powder particles suitable for mosquito-driven dissemination" - Is there a chance that a liquid formulation would work? Couldn't such a formulation also be impregnated into the female mosquito?

Liquid suspensions are used in modified bottle bioassays, where the inner surface of bottles are covered with the suspension and then left to dry so that a layer of powder remains on the surface; this, however, seems unlikely to work for wider dissemination under field conditions.

- lines 83-84, "(iv) have mechanisms of action that differ from that of the insecticide targeted for management" - I didn't understand. Wouldn't this strategy be part of management? Or is the text referring to 'adulticides targeted for management'?

Thanks for raising this point; we have rephrased this confusing bit to "(iv) have mechanisms of action that differ from that of the insecticide towards which mosquitoes have evolved resistance" (l. 86).

Materials and methods

Item 'Diagnostic-concentration bioassays' (starts on line 120): as already mentioned, I suggest providing more details about the monitoring time, or the average mortality time, of larvae exposed to each product.

As mentioned in our previous response to this point, we followed all larvae/pupae until emergence or death (l. 140-143 and 164), but did not record the time to event in every individual juvenile. This was only done for studying the lifespan of adult females.

- Line 155, "Adult mosquitoes remained inside the cages until death or for up to 60 days." - inform that (if) the females received any source of sugar as food during this period.

Yes, they did (l. 152-153); we apologise for not having mentioned this in the first version.

Results

- Item 'Diagnostic-concentration bioassays', lines 178-183 - It would be interesting to include information on the time dynamics, or mean time, of mortality with each product tested, since their modes of action are distinct. DFB is supposed to impact each molt, while the expected action of SPN, a neurotoxic agent, is quick. Even in the standard assays recommended for these products, there are differences in follow-up time (generally 24 hours for neurotoxic agents and several days for IGR ones).

As before, did not record the time to emergence/death for every individual juvenile in these bioassays. We did record, however, whether death occurred at the larval or pupal stage – as shown, e.g., in Fig. 3 (for bioassays) or Table I (for cage experiments). See also l. 186-187 and 197-199.

Discussion

Lines 247-248, "DFB thus emerges as a potentially useful alternative or complement to PPF for mosquito-driven dissemination" - insecticide rotation in order to preserve them.

Thanks once again for the suggestion; we now mention this point in l. 255-257.

Line 257, reference to "Because we found some evidence suggesting leakage of PPF" - Judging by Figure 4, leakage appears to have occurred for all three products tested. To what factor do the authors attribute this effect? Some technical problem with any undesirable 'opening' of the 10 'closed cups' that allowed adult females to enter? Or to the spread of the products in the cage environment by females during flight?

Evidence for leakage is, as we discuss, much clearer for PPF than for DFB (and, we note, nonexistent for SPN; see Fig. 4 and lower row of Fig. 5, as well as the 'SPN' coefficient in Table II). We suspect that some of the tiny particles did somehow find their way from the air within test cages into some of the closed cups – perhaps through tissue pores, or perhaps when the cover was removed... We feel, however, that discussing possible mechanisms for this observation would be too speculative, and prefer to keep our brief discussion as is. We nevertheless now include some comments on leakage in a new paragraph on study limitations (see l. 307-311).

Lines 285-286, "Cage tests should proceed in parallel to investigate the potential usefulness of combining DFB with PPF" – It's worth considering (and discussing) that this field initiative involving the simultaneous use of two IGRs has the potential to accelerate the development and spread of resistance to both compounds. It's a way to prioritize chemical control. While it may have a significant effect in the short term, it risks compromising more sustainable strategies, in which chemical vector control should be used as a complement to vector control.

We thank the Reviewer for raising this point. After careful consideration, we have decided to remove the sentence highlighted by the Reviewer.

A curiosity: it would be pertinent to include, in the Discussion, information on the average effectiveness of MD-PPF (in terms of adult emergence inhibition) currently in field use.

Thank you for the suggestion. We now mention (l. 270-271) that our estimate of PPF effects on adult emergence (30.4% predicted emergence; 95%CI, 20.0–43.3%) are "overall consistent with the results of previous field trials" (refs. 13-16).

Additionally, it is worth discussing, based on (a) the performance, in the "simulated field" trials, of each of the insecticides presented in this Manuscript, and (b) the effectiveness of MD-PPF already in field use, whether it is possible to estimate, or envision, the effect of implementing MD with other products (mainly DFB) in routine control.

As with the point of leakage into closed cups above, we feel that it is too early to speculate about the potential of MD-DFB to reach routine control. As we mention in the manuscript (e.g., l. 46, 312-319), we will first need to see if it works in small-scale field trials (with entomological endpoints) – and then, based on those results, decide whether it seems reasonable to run larger-scale trials with epidemiological endpoints. We now stress external validity issues in a new paragraph on study limitations (see l. 294-299).

Minor comments

Table I, legend, "(from disposable lab coats)" – This is already included in the methodology. There's no need to repeat it

We prefer to keep this small detail so that the legend is self-explanatory.

Discussion, line 258, "DFB (Fig. 2, Table II)" - or Figure 4?

Corrected, thanks.

## REVIEWER #3

a) Adequacy of the abstract; b) Originality and importance of the contribution for the development of the field of study. And relevance of: c) Methodology, results and discussion; d) References; e) Figures and tables.

The correspondence we received contained no comments from Reviewer #3.

## SECOND REVIEW ROUND

REVIEWERS' COMMENTS

## REVIEWER #1

The authors took all my previous comments into consideration, addressing or justifying each point. The manuscript has improved in consistency and argumentation.

I have only two additional comments:

- in the Introduction, regarding lines 84-85: "(iii) are formulated as powder particles suitable for mosquito-driven dissemination." In the previous version, I questioned the feasibility of using liquid formulations in dissemination stations. There is a publication in this journal, MIOC, that addresses this topic, and it would be pertinent to consider:

https://www.scielo.br/j/mioc/a/TtV9btYbySrb7yvnDgMmnFc/?lang=en

- in the Discussion, regarding the previous reference to the section "Because we found some evidence suggesting leakage of PPF," I thank the authors for their response and for including the possibility of leakage also in samples with DFB. Just for clarification, I was comparing, in Figure 4, the closed and open cups of each product

tested. In this figure, judging by the confidence intervals presented, there is a difference (closed vs. open) in the three products—even for SPN. Although, evidently, this difference (closed vs. open cups) is much smaller for SPN than for the other two products, the confidence intervals between the two types of cups do not overlap in this experimental condition. It was on this observation that I based my previous comment.

- By the way, still regarding Figure 4, it is worth noting that there is a difference, for PPF, between the value for the closed cups shown in the legend (0.78) and the value in the graph (below 0.75). It is likely that the graph is showing the result including the closed cup that was suspected of contamination (excluded from the value shown in the legend). However, to avoid confusion in interpretation, it might be interesting to make this point in the legend more explicitly, or perhaps (?) to put both results on the graph, considering and not considering the 'contaminated cup'.

**REVIEWER #2**

Authors have addressed the concerns point by Reviewers.

