## [Reviewer Report · FIRST REVIEW ROUND - REVIEWERS COMMENTS]

## Reviewer #1

Dear Editor,

This study set out to evaluate alternative larvicides for mosquito-driven dissemination, given the increasing risk of resistance to pyriproxyfen (PPF), a juvenile hormone analogue. Through simulated, blinded, and controlled cage experiments, the authors assessed the potential for Aedes aegypti females to disseminate diflubenzuron (DFB) and Spinosad (SPN), in comparison to the gold standard, PPF. The results showed that both DFB and, to a lesser extent, SPN significantly reduced adult mosquito emergence in untreated containers, though less effectively than PPF. Notably, exposure to SPN rapidly reduced adult female survival, likely limiting its potential for dissemination. The authors conclude that diflubenzuron could be a promising alternative to PPF and merits further field testing.

The manuscript is well written and addresses a timely and relevant topic, particularly given the limited number of alternative active ingredients suitable for mosquito autodissemination - a strategy the WHO considers a promising complementary tool for reducing vector populations and arbovirus transmission. The statistical analyses appear robust. However, there are several concerns that should be addressed prior to publication.

1. Rationale for insecticide selection

A core principle of the autodissemination strategy is the use of slow-acting compounds that allow female mosquitoes to survive long enough to transfer the insecticide to larval habitats. Additionally, the target mosquito population should be susceptible to the insecticide deployed at the dissemination station to minimize resistance selection and ensure the long-term efficacy of the strategy.

These considerations raise questions about the suitability of the two alternative products tested in this study:

• Spinosad (SPN) is a biopesticide with strong adulticidal activity, as confirmed by the authors and previously demonstrated in multiple studies. Given its fast-acting nature, what was the rationale for selecting SPN for a dissemination strategy, knowing that it would likely kill females before they could transfer the product to breeding sites?

• Diflubenzuron (DFB), like PPF, is an insect growth regulator (IGR) that inhibits chitin synthesis. Moderate levels of resistance to both PPF and DFB have been documented in Ae. aegypti populations, including in Brazil (Campos et al. 2020, 2023; Araújo et al. 2019). P450 monooxygenases have been implicated in PPF detoxification, and cross-resistance due to elevated P450 activity is plausible (Yunta et al. 2016). Given the similar mode of action and potential for cross-resistance, what is the justification for selecting DFB as a viable alternative for resistance management?

Would it not be more appropriate to test new compounds with unrelated modes of action to PPF, such as slow-acting pyrroles (e.g., chlorfenapyr), oxadiazines, or some neonicotinoids (e.g., imidacloprid), possibly in combination, to enhance efficacy and mitigate resistance? his This a key point, especially since the authors justified the rationale of the study within the framework of integrated vector management.

2. Resistance diagnostics

The section on diagnostic concentrations (DC) requires clarification. According to WHO guidelines (WHO, 2022), DCs should be established using a susceptible laboratory reference strain (e.g., Rockefeller, New Orleans, Bora Bora). If the same field-collected population was used both to define DCs and to assess resistance levels, the results may be unreliable or misleading. This section should be revised accordingly or removed if appropriate data are lacking.

Minor comments and clarifications:

Introduction

• Line 96: Consider rephrasing as “Our study showed that diflubenzuron...”

• Line 90: WHOPES has been replaced by WHO PQT since 2017. Please update the terminology and provide a link to the list of WHO-prequalified vector control products.

Materials and Methods

• Line 100: What was the age of the adult females used in the bioassays? Why not use a standard laboratory colony (e.g., Rockefeller) to minimize variability due to genetic background?

Mosquito-driven dissemination: cage experiments

The autodissemination station is insufficiently described. The use of plastic cups is mentioned, but the design should be detailed in the main text. Reference should be made to Figures 1 and 2 for clarity.

Results: diagnostic concentration bioassays

• If no susceptible reference strain was used to establish DCs, the methodology and conclusions on susceptibility may be invalid. Please clarify this point and revise the section if necessary.

Results

• Line 189: “Sharp decline in observed adult emergence” – please quantify the reduction and specify if it was statistically significant.

• Line 209: How do the authors explain the relatively high mortality in control cups (13%)? Could cross-contamination have occurred in the controls, similar to what has been reported for PPF?

• Despite a relatively large sample size (n = 407 over two rounds), DFB’s impact on emergence was not statistically significant (47% emergence vs. 87% in controls). Could there be any bias or lack of power in the statistical model? Have you test other models?

Discussion

• L249: The authors indicate a significant reduction in precited emergence with DFB, whereas Table 2 suggests the reduction is not statistically significant (Estimate = –0.937, P = 0.1138). Please clarify this discrepancy.

• The rise of Aedes resistance to insect growth regulators (IGRs) such as PPF and DFB in Brazil should be discussed, as it may threaten the future success of the autodissemination strategy based on DFB.

References:

1. de Araújo AP, et al. (2019). J Insect Sci. 19(3):16. [PMCID: PMC6556078]

2. Campos KB, et al. (2020). Parasit Vectors. 13(1):531. [PMCID: PMC7590490]

3. Campos KB, et al. (2023). J Vector Ecol. 48(1):12–1

4. Yunta C, et al. (2016). Insect Biochem Mol Biol. 78:50–57. [PMCID: PMC6399515]

5. WHO 2022. Manual for monitoring insecticide resistance in mosquito vectors and selecting appropriate interventions, 22 June 2022, Geneva

## Reviewer #2

Insecticide dissemination stations, implemented relatively recently to control Aedes aegypti in Brazil, are based on the characteristic “jumping posture” of this species’ females. The rationale behind this method is to ensure the spread of larvicides across different potential mosquito breeding sites by the adult females themselves.

The manuscript reports on “simulated” trials in large cages on the feasibility of using two insecticides not previously tested using this methodology with the dissemination stations: diflubenzuron, a chitin synthesis inhibitor, and spinosad, a neurotoxic insecticide that modulates acetylcholine receptors. The trials are well designed, executed, and analyzed. The team has extensive experience in developing and implementing this methodology and even includes professionals responsible for establishing “dissemination stations” for Aedes aegypti control in Brazil.

Furthermore, validation for use of alternative larvicides to pyriproxyfen (a juvenile hormone analogue) in dissemination stations is welcome, as it helps to identify A. aegypti chemical control options. It is worth noting that one of the Ministry of Health (MoH) guidelines is to encourage the rotation of products with different mechanisms of action, in order to prevent, or to delay, the spread of vector resistance, thus preserving the products used in the chemical control of Aedes aegypti.

However, I understand that the main shortcoming of the work is the lack of a more detailed technical justification and argument for the inclusion of spinosad in the tests. Diflubenzuron and pyriproxyfen interfere with physiological, or endocrinological, processes specific to the immature stages. In contrast, spinosad has the potential to also act on adults; moreover, unlike the other two products, because it is a neurotoxic agent, its action is rapid, potentially leading to the death of the adult female, precisely the specimen that performs the “dissemination work.” This conclusion, in fact, was confirmed by the results of the manuscript. Exploring the mode of action of each insecticide in the Introduction, anticipating the expected outcomes of the trials, could be a way to add robustness to the text. Furthermore, detailing, in the methodology and/or results, the time employed to monitor the larvae treated with each of the different products helps to highlight the differences in their modes of action.

Below, observations in the order they appear in the text.

Abstract, “Alternative larvicide/pupicide molecules suitable for mosquito-driven dissemination, but with distinct modes of action, are therefore needed.”- It is worth mentioning (in this section or in the Introduction) that this initiative is in line with the recommendations of the MoH, which advocate rotation of products with different modes of action, as a way of ensuring greater longevity of the available insecticides, preserving them.

Introduction

I suggest commenting in this section that dissemination stations are successful in controlling Aedes aegypti because they exploit two unique characteristics of adult females of this species: (1) they spread their eggs across many breeding sites, and (2) they prefer artificial containers, either inside or outside home, used by humans.

- on the requirements for insecticides to work in dissemination stations:

- lines 82-83, “(iii) are formulated as powder particles suitable for mosquito-driven dissemination” - Is there a chance that a liquid formulation would work? Couldn’t such a formulation also be impregnated into the female mosquito?

- lines 83-84, “(iv) have mechanisms of action that differ from that of the insecticide targeted for management” - I didn’t understand. Wouldn’t this strategy be part of management? Or is the text referring to ‘adulticides targeted for management’?

Materials and methods

Item ‘Diagnostic-concentration bioassays’ (starts on line 120): as already mentioned, I suggest providing more details about the monitoring time, or the average mortality time, of larvae exposed to each product.

– Line 155, “Adult mosquitoes remained inside the cages until death or for up to 60 days.” - inform that (if) the females received any source of sugar as food during this period.

Results

- Item ‘Diagnostic-concentration bioassays’, lines 178-183 - It would be interesting to include information on the time dynamics, or mean time, of mortality with each product tested, since their modes of action are distinct. DFB is supposed to impact each molt, while the expected action of SPN, a neurotoxic agent, is quick. Even in the standard assays recommended for these products, there are differences in follow-up time (generally 24 hours for neurotoxic agents and several days for IGR ones).

Discussion

Lines 247-248, “DFB thus emerges as a potentially useful alternative or complement to PPF for mosquito-driven dissemination” - insecticide rotation in order to preserve them.

Line 257, reference to “Because we found some evidence suggesting leakage of PPF” - Judging by Figure 4, leakage appears to have occurred for all three products tested. To what factor do the authors attribute this effect? Some technical problem with any undesirable ‘opening’ of the ‘closed cups’ that allowed adult females to enter? Or to the spread of the products in the cage environment by females during flight?

Lines 285-286, “Cage tests should proceed in parallel to investigate the potential usefulness of combining DFB with PPF” – It’s worth considering (and discussing) that this field initiative involving the simultaneous use of two IGRs has the potential to accelerate the development and spread of resistance to both compounds. It’s a way to prioritize chemical control. While it may have a significant effect in the short term, it risks compromising more sustainable strategies, in which chemical vector control should be used as a complement to vector control.

A curiosity: it would be pertinent to include, in the Discussion, information on the average effectiveness of MD-PPF (in terms of adult emergence inhibition) currently in field use. Additionally, it is worth discussing, based on (a) the performance, in the “simulated field” trials, of each of the insecticides presented in this Manuscript, and (b) the effectiveness of MD-PPF already in field use, whether it is possible to estimate, or envision, the effect of implementing MD with other products (mainly DFB) in routine control.

Minor comments

Table I, legend, “(from disposable lab coats)” – This is already included in the methodology. There’s no need to repeat it

Discussion, line 258, “DFB (Fig. 2, Table II)” - or Figure 4?

## Reviewer #3

a) Adequacy of the abstract; b) Originality and importance of the contribution for the development of the field of study. And relevance of: c) Methodology, results and discussion; d) References; e) Figures and tables.

---

## [Author Response · AUTHORS RESPONSE TO REVIEWERS]

## Belo Horizonte, 20/08/2025

Dear Dr Martins,

we first want to thank you and the Reviewers for the detailed, overall positive assessment of our work, for the constructive criticism, and for the useful comments and suggestions.

In particular, we appreciate the opportunity to clarify the rationale for testing spinosad – a point that, we agree, was not well explained in the first version of the manuscript. In short (see also below and R1 version), we wanted to test if, and to what extent, spinosad shortened adult lifespan in the specific context of autodissemination. The idea was that, if spinosad killed females only slowly (too slowly to significantly hinder dissemination), then it could hold promise for autodissemination – similar to (but simpler than) the use of slow-killing entomopathogenic fungi in combination with PPF in autodissemination stations.

We also take the opportunity to clarify that we did not experimentally define bioassay Diagnostic Concentrations (DCs). This is because, with that part of the study, we just aimed at making sure that (1) our mosquitoes were not resistant to the active ingredients we used and (2) those ingredients were indeed active and working as expected. For that, we simply retrieved DCs from the literature – as indicated in l. 128-129: “...based on published results, we used 0.02 parts per million (ppm) for PPF, 1.5 ppm for DFB, and 1 ppm for SPN.(29-31)...”, where the refs. are:

29. Andrighetti MTM, Cerone F, Rigueti M, Galvani KC, Macoris MLG. Effect of pyriproxyfen in Aedes aegypti populations with different levels of susceptibility to the organophosphate temephos. Dengue Bull. 2008; 32: 186-98.

30. Montaño-Reyes A, Llanderal-Cazares C, Valdez-Carrasco J, Miranda-Perkins K, Sanchez-Arroyo H. Susceptibility and alterations by diflubenzuron in larvae of Aedes aegypti. Arch Insect Biochem Physiol. 2019; 102: e21604.

31. Aldridge RL, Alto BW, Connelly CR, Okech B, Siegfried B, Linthicum KJ. Lethal and sublethal concentrations of formulated larvicides against susceptible Aedes aegypti. J Am Mosq Control Assoc. 2022; 38: 250-60.

We have added a full paragraph (l. 294-311) explicitly covering the main limitations of the study, including the point on DCs. Finally, we have addressed each of the points raised by the Reviewers. Below you can find, in red font, our point-by-point responses – with line numbers for any changes made to the R1 manuscript, in which all such changes are highlighted in yellow.

We hope that you will find that this revised (and clearly improved, we believe) version of our manuscript meets the standards of Memórias, and look forward to hearing from you.

Sincerely,

Fernando Abad-Franch (on behalf of all co-authors)

---

## [Reviewer Report · REVIEWERS COMMENTS]

## Reviewer #1

Dear Editor,

This study set out to evaluate alternative larvicides for mosquito-driven dissemination, given the increasing risk of resistance to pyriproxyfen (PPF), a juvenile hormone analogue. Through simulated, blinded, and controlled cage experiments, the authors assessed the potential for Aedes aegypti females to disseminate diflubenzuron (DFB) and Spinosad (SPN), in comparison to the gold standard, PPF. The results showed that both DFB and, to a lesser extent, SPN significantly reduced adult mosquito emergence in untreated containers, though less effectively than PPF. Notably, exposure to SPN rapidly reduced adult female survival, likely limiting its potential for dissemination. The authors conclude that diflubenzuron could be a promising alternative to PPF and merits further field testing.

The manuscript is well written and addresses a timely and relevant topic, particularly given the limited number of alternative active ingredients suitable for mosquito autodissemination – a strategy the WHO considers a promising complementary tool for reducing vector populations and arbovirus transmission. The statistical analyses appear robust. However, there are several concerns that should be addressed prior to publication.

We thank the Reviewer for their clear summary of our work and positive comments.

1. Rationale for insecticide selection A core principle of the autodissemination strategy is the use of slow-acting compounds that allow female mosquitoes to survive long enough to transfer the insecticide to larval habitats. Additionally, the target mosquito population should be susceptible to the insecticide deployed at the dissemination station to minimize resistance selection and ensure the long-term efficacy of the strategy.

These considerations raise questions about the suitability of the two alternative products tested in this study:

• Spinosad (SPN) is a biopesticide with strong adulticidal activity, as confirmed by the authors and previously demonstrated in multiple studies. Given its fast-acting nature, what was the rationale for selecting SPN for a dissemination strategy, knowing that it would likely kill females before they could transfer the product to breeding sites?

This is an important point that we agree we did not explain clearly in our original submission. We first wanted to test if, and to what extent, SPN was capable of shortening adult lifespan in the specific context of autodissemination – i.e., after ‘non-forced’ contact with dust-treated surfaces, thus mimicking ‘real-life’ conditions in which mosquitoes land and possibly walk on the walls of a potential breeding site. The idea was that, if spinosad killed females only slowly 3 (too slowly to significantly hinder dissemination), then it could hold promise for autodissemination – similar to (but simpler than) the use of slow-killing entomopathogenic fungi in combination with PPF in autodissemination stations (cf. ref. 19). We found out that this is not the case – and, as a ‘side effect’, demonstrated that spinosad might be interesting for use in lethal ovitraps (l. 290-291). We have tried to clarify the rationale for our choice of testing spinosad (l. 98-100, 224-225, and 283-284; see also l. 230-231, 254-255, 290-293).

• Diflubenzuron (DFB), like PPF, is an insect growth regulator (IGR) that inhibits chitin synthesis. Moderate levels of resistance to both PPF and DFB have been documented in Ae. aegypti populations, including in Brazil (Campos et al. 2020, 2023; Araújo et al. 2019). P450 monooxygenases have been implicated in PPF detoxification, and cross-resistance due to elevated P450 activity is plausible (Yunta et al. 2016). Given the similar mode of action and potential for cross-resistance, what is the justification for selecting DFB as a viable alternative for resistance management?

While we are aware that detoxification of DFB and PPF may involve some common biochemical pathways, we:

(a) Note that DFB and PPF have distinct modes of action (as defined by the Insecticide Resistance Action Committee – IRAC; see https://irac-online.org/mode-of-action/); and

(b) Know of no evidence of DFB/PPF cross-resistance in Aedes.

All in all, even if some level of cross-resistance happens to emerge (a possibility we now stress), having two molecules with distinct modes of action (as defined by IRAC) available for use is still a clear advantage. We have clarified these points in l. 257-262 (and added the new refs. 43,44).

Would it not be more appropriate to test new compounds with unrelated modes of action to PPF, such as slow-acting pyrroles (e.g., chlorfenapyr), oxadiazines, or some neonicotinoids (e.g., imidacloprid), possibly in combination, to enhance efficacy and mitigate resistance? This is a key point, especially since the authors justified the rationale of the study within the framework of integrated vector management.

We are indeed considering and testing further compounds – with an eye on those that are approved for use in drinking water, act at low doses, and are formulated as (or can be micronized to) fine powder suitable for dissemination by mosquitoes (l. 80-89 and 245-246). We mention a few of them (l. 319-322), but this is currently work in progress.

2. Resistance diagnostics

The section on diagnostic concentrations (DC) requires clarification. According to WHO guidelines (WHO, 2022), DCs should be established using a susceptible laboratory reference strain (e.g., Rockefeller, New Orleans, Bora Bora). If the same field-collected population was used both to define DCs and to assess resistance levels, the results may be unreliable or misleading. This section should be revised accordingly or removed if appropriate data are lacking.

Please note that the only aim of this part of the study was to confirm that the products available to us indeed killed mosquitoes from our colony. We did not aim at estimating, and did not estimate, our bioassay DCs – we simply retrieved them from the literature (refs. 29-31), as stated explicitly in l. 127-129.

Minor comments and clarifications:

Introduction

● Line 96: Consider rephrasing as “Our study showed that diflubenzuron...”

We have rephrased the end of the Introduction (please see l. 98-102).

● Line 90: WHOPES has been replaced by WHO PQT since 2017. Please update the terminology and provide a link to the list of WHO-prequalified vector control products.

Many thanks for this suggestion; we have now updated the terminology and provide specific refs. and links (refs. 24-27).

Materials and Methods

● Line 100: What was the age of the adult females used in the bioassays? Why not use a standard laboratory colony (e.g., Rockefeller) to minimize variability due to genetic background?

Females were 7-10 days old (see l. 151-152). We used field-derived females because we wanted to know if our test compounds worked with local populations from the sites where we conduct small-scale field tests (e.g., ref. 15); we now clarify this (l. 296-299).

Mosquito-driven dissemination: cage experiments

● The autodissemination station is insufficiently described. The use of plastic cups is mentioned, but the design should be detailed in the main text. Reference should be made to Figures 1 and 2 for clarity.

We now make explicit reference to Figs. 1 and 2 and refer readers to refs. 13 and 15 for a more detailed description of DSs (l. 150-155).

Results: diagnostic concentration bioassays

● If no susceptible reference strain was used to establish DCs, the methodology and conclusions on susceptibility may be invalid. Please clarify this point and revise the section if necessary.

Please see our response on this point above. To reiterate, the only aim of this part of the study was to confirm that the products available to us indeed killed mosquitoes from our colony. We did not aim at estimating, and did not estimate, our bioassay DCs – we simply retrieved them from the literature (refs. 29-31), as stated explicitly in l. 127-129.

Results

• Line 189: “Sharp decline in observed adult emergence” – please quantify the reduction and specify if it was statistically significant.

Please note that the numerical results are given in Table I (observed values) and Table II (numerical output from the top-performing GLMM, including estimates, standard errors, zstatistics, p-values, and 95% CI limits). Fig. 4 shows observed proportions (with score 95% CIs), and Fig. 5 shows predictions (on the probability scale) from the top-performing GLMM, with 95% CIs. We can include a summary of those results in the text, but this might seem redundant.

• Line 209: How do the authors explain the relatively high mortality in control cups (13%)? Could cross-contamination have occurred in the controls, similar to what has been reported for PPF?

We think it unlikely that any of the open cups in control cages might have become contaminated – none of them had the sharp increase in juvenile mortality we’d expect if that were the case. Mortality in closed cups within control cages was ~10%, and the small increase (to ~13%) in open cups can be parsimoniously attributed to higher crowding in open cups – where females laid eggs which developed into larvae. In particular, the highest mortality in an open cup set within a control cage was 33.3%, and this was the open cup with the largest number of larvae (57, of which 18 died as larvae and 1 as a pupa). Excluding this cup, the average mortality was identical (10%) to that seen in closed cups within control cages.

• Despite a relatively large sample size (n = 407 over two rounds), DFB’s impact on emergence was not statistically significant (47% emergence vs. 87% in controls). Could there be any bias or lack of power in the statistical model?

Have you test other models?

We follow leading statisticians (see, e.g., among many recent relevant papers, https://www.nature.com/articles/d41586-019-00857-9; https://peerj.com/articles/3544/; https://www.nature.com/articles/s41562-017-0224-0 – and some older ones, e.g., Cohen’s classic https://users.cla.umn.edu/~nwaller/prelim/cohenearthround.pdf) in avoiding the use of ‘statistical significance’ to learn from our data. We believe that a reduction from 90% to less than 56% (the GLMM estimates most compatible with our data, at mean juvenile crowding) is biologically significant – and perhaps also significant in practice, although this remains to be tested in real-world scenarios (l. 45-46 and 312-319). Please see also our next response.

Discussion

• L249: The authors indicate a significant reduction in precited emergence with DFB, whereas Table 2 suggests the reduction is not statistically significant (Estimate = –0.937, P = 0.1138). Please clarify this discrepancy.

We meant significant in the ordinary (not ‘statistical’) sense of the word. This was because we believe that reducing emergence from ~90% (control cages) to ~56% (with an upper 95%CI limit of 69%) is ‘significant’. To avoid confusion, however, we have removed “significant” from that sentence (l. 263).

We note, in any case, that the coefficient and p-value the Reviewer highlights are for the effect of DFB in closed cups – the main effect in a model with an interaction between cup type (closed, open) and treatment (CTR, DFB, SPN, PPF). The coefficient for the interaction (-1.059, SE 0.594 is associated with a much smaller p-value of 0.074). The ‘effects’ of both coefficients (main effect, interaction), together with that of the ‘cup type’ and ‘crowding’ main effects (and of course the intercept), yield the predictions we highlight – which we do precisely because interpreting predictions on a natural scale (here, %) is usually much more intuitive than interpreting (link-scale) model coefficients, especially in the presence of interactions.

(Note also that the interaction coefficient estimates the net effect of DFB in open cups; exponentiating the point estimate and 95%CI limits, our best estimate is that DFB caused a reduction of ~65.3% in the odds of emergence, with the CI spanning from a ~89.2% reduction to an 11.0% increase, relative to the difference between closed and open cups in the CTR treatment – where the model estimates a ~14.8% increase in the odds of emergence in open compared to closed cups [where the model predicts emergence at exp(2.189)/(1+(exp(2.189))) 0.89926, or ~89.9%, when juvenile crowding is ~28 per 100 mL of water].)

• The rise of Aedes resistance to insect growth regulators (IGRs) such as PPF and DFB in Brazil should be discussed, as it may threaten the future success of the autodissemination strategy based on DFB.

As mentioned above, we now discuss both resistance to individual compounds and the possibility of cross-resistance (see l. 257-262).

References:

1. de Araújo AP, et al. (2019). J Insect Sci. 19(3):16. [PMCID: PMC6556078]

2. Campos KB, et al. (2020). Parasit Vectors. 13(1):531. [PMCID: PMC7590490]

3. Campos KB, et al. (2023). J Vector Ecol. 48(1):12–1

4. Yunta C, et al. (2016). Insect Biochem Mol Biol. 78:50–57. [PMCID: PMC6399515]

5. WHO 2022. Manual for monitoring insecticide resistance in mosquito vectors and selecting appropriate interventions, 22 June 2022, Geneva.

## Reviewer #2

Insecticide dissemination stations, implemented relatively recently to control Aedes aegypti in Brazil, are based on the characteristic “jumping posture” of this species’ females. The rationale behind this method is to ensure the spread of larvicides across different potential mosquito breeding sites by the adult females themselves.

The manuscript reports on “simulated” trials in large cages on the feasibility of using two insecticides not previously tested using this methodology with the dissemination stations: diflubenzuron, a chitin synthesis inhibitor, and spinosad, a neurotoxic insecticide that modulates acetylcholine receptors. The trials are well designed, executed, and analyzed. The team has extensive experience in developing and implementing this methodology and even includes professionals responsible for establishing “dissemination stations” for Aedes aegypti control in Brazil.

Furthermore, validation for use of alternative larvicides to pyriproxyfen (a juvenile hormone analogue) in dissemination stations is welcome, as it helps to identify A. aegypti chemical control options. It is worth noting that one of the Ministry of Health (MoH) guidelines is to encourage the rotation of products with different mechanisms of action, in order to prevent, or to delay, the spread of vector resistance, thus preserving the products used in the chemical control of Aedes aegypti.

We also thank Reviewer #2 for their positive remarks on our report.

However, I understand that the main shortcoming of the work is the lack of a more detailed technical justification and argument for the inclusion of spinosad in the tests. Diflubenzuron and pyriproxyfen interfere with physiological, or endocrinological, processes specific to the immature stages. In contrast, spinosad has the potential to also act on adults; moreover, unlike the other two products, because it is a neurotoxic agent, its action is rapid, potentially leading to the death of the adult female, precisely the specimen that performs the “dissemination work.” This conclusion, in fact, was confirmed by the results of the manuscript. Exploring the mode of action of each insecticide in the Introduction, anticipating the expected outcomes of the trials, could be a way to add robustness to the text. Furthermore, detailing, in the methodology and/or results, the time employed to monitor the larvae treated with each of the different products helps to highlight the differences in their modes of action.

This is an important point that was also raised by Reviewer #1 and that, as discussed in our response to their comments, we agree we did not explain clearly in our original submission. Our response to Reviewer #1 reads: “We first wanted to test if, and to what extent, SPN was capable of shortening adult lifespan in the specific context of autodissemination – i.e., after ‘non-forced’ contact with dust-treated surfaces, thus mimicking ‘real-life’ conditions in which mosquitoes land and possibly walk on the walls of a potential breeding site. The idea was that, if spinosad killed females only slowly (too slowly to significantly hinder dissemination), then it could hold promise for autodissemination – similar to (but simpler than) the use of slow-killing 8 entomopathogenic fungi in combination with PPF in autodissemination stations (cf. ref. 19). We found out that this is not the case – and, as a ‘side effect’, demonstrated that spinosad might be interesting for use in lethal ovitraps (l. 290-291). We have tried to clarify the rationale for our choice of testing spinosad (l. 98-100, 224-225, and 283-284; see also l. 230-231, 254-255, 290- 293).”

We followed all larvae/pupae until emergence or death (l. 140-143 and 164), but did not record the time to event in every individual juvenile. This was only done for studying the lifespan of adult females.

Below, observations in the order they appear in the text.

Abstract, “Alternative larvicide/pupicide molecules suitable for mosquito-driven dissemination, but with distinct modes of action, are therefore needed.”- It is worth mentioning (in this section or in the Introduction) that this initiative is in line with the recommendations of the MoH, which advocate rotation of products with different modes of action, as a way of ensuring greater longevity of the available insecticides, preserving them.

The recommendations by the Brazilian MoH are only available in Portuguese (the most recent, at https://www.gov.br/saude/pt-br/centrais-de-conteudo/publicacoes/estudos-e-notasinformativas/2024/nota-informativa-no-29-2024-cgarb-dedt-svsa-ms). We think that refs. 20- 23, and especially ref. 22 (https://www.nature.com/articles/s41598-021-03367-9.pdf) provide a much more general and widely accessible overview on insecticide resistance and its management – including the importance of “the availability of active ingredients with diverse modes of action and that can be deployed safely and conveniently at scale” (l. 79-80).

Introduction

I suggest commenting in this section that dissemination stations are successful in controlling Aedes aegypti because they exploit two unique characteristics of adult females of this species: (1) they spread their eggs across many breeding sites, and (2) they prefer artificial containers, either inside or outside home, used by humans.

Thank you for this suggestion; we now mention those characteristics in l. 71-72.

- on the requirements for insecticides to work in dissemination stations:

- lines 82-83, “(iii) are formulated as powder particles suitable for mosquito-driven dissemination” - Is there a chance that a liquid formulation would work? Couldn’t such a formulation also be impregnated into the female mosquito?

Liquid suspensions are used in modified bottle bioassays, where the inner surface of bottles are covered with the suspension and then left to dry so that a layer of powder remains on the surface; this, however, seems unlikely to work for wider dissemination under field conditions.

- lines 83-84, “(iv) have mechanisms of action that differ from that of the insecticide targeted for management” - I didn’t understand. Wouldn’t this strategy be part of management? Or is the text referring to ‘adulticides targeted for management’?

Thanks for raising this point; we have rephrased this confusing bit to “(iv) have mechanisms of action that differ from that of the insecticide towards which mosquitoes have evolved resistance” (l. 86).

Materials and methods

Item ‘Diagnostic-concentration bioassays’ (starts on line 120): as already mentioned, I suggest providing more details about the monitoring time, or the average mortality time, of larvae exposed to each product.

As mentioned in our previous response to this point, we followed all larvae/pupae until emergence or death (l. 140-143 and 164), but did not record the time to event in every individual juvenile. This was only done for studying the lifespan of adult females.

- Line 155, “Adult mosquitoes remained inside the cages until death or for up to 60 days.” - inform that (if) the females received any source of sugar as food during this period.

Yes, they did (l. 152-153); we apologise for not having mentioned this in the first version.

Results

- Item ‘Diagnostic-concentration bioassays’, lines 178-183 - It would be interesting to include information on the time dynamics, or mean time, of mortality with each product tested, since their modes of action are distinct. DFB is supposed to impact each molt, while the expected action of SPN, a neurotoxic agent, is quick. Even in the standard assays recommended for these products, there are differences in follow-up time (generally 24 hours for neurotoxic agents and several days for IGR ones).

As before, did not record the time to emergence/death for every individual juvenile in these bioassays. We did record, however, whether death occurred at the larval or pupal stage – as shown, e.g., in Fig. 3 (for bioassays) or Table I (for cage experiments). See also l. 186-187 and 197-199.

Discussion

Lines 247-248, “DFB thus emerges as a potentially useful alternative or complement to PPF for mosquito-driven dissemination” - insecticide rotation in order to preserve them.

Thanks once again for the suggestion; we now mention this point in l. 255-257.

Line 257, reference to “Because we found some evidence suggesting leakage of PPF” - Judging by Figure 4, leakage appears to have occurred for all three products tested. To what factor do the authors attribute this effect? Some technical problem with any undesirable ‘opening’ of the 10 ‘closed cups’ that allowed adult females to enter? Or to the spread of the products in the cage environment by females during flight?

Evidence for leakage is, as we discuss, much clearer for PPF than for DFB (and, we note, nonexistent for SPN; see Fig. 4 and lower row of Fig. 5, as well as the ‘SPN’ coefficient in Table II). We suspect that some of the tiny particles did somehow find their way from the air within test cages into some of the closed cups – perhaps through tissue pores, or perhaps when the cover was removed... We feel, however, that discussing possible mechanisms for this observation would be too speculative, and prefer to keep our brief discussion as is. We nevertheless now include some comments on leakage in a new paragraph on study limitations (see l. 307-311).

Lines 285-286, “Cage tests should proceed in parallel to investigate the potential usefulness of combining DFB with PPF” – It’s worth considering (and discussing) that this field initiative involving the simultaneous use of two IGRs has the potential to accelerate the development and spread of resistance to both compounds. It’s a way to prioritize chemical control. While it may have a significant effect in the short term, it risks compromising more sustainable strategies, in which chemical vector control should be used as a complement to vector control.

We thank the Reviewer for raising this point. After careful consideration, we have decided to remove the sentence highlighted by the Reviewer.

A curiosity: it would be pertinent to include, in the Discussion, information on the average effectiveness of MD-PPF (in terms of adult emergence inhibition) currently in field use.

Thank you for the suggestion. We now mention (l. 270-271) that our estimate of PPF effects on adult emergence (30.4% predicted emergence; 95%CI, 20.0–43.3%) are “overall consistent with the results of previous field trials” (refs. 13-16).

Additionally, it is worth discussing, based on (a) the performance, in the “simulated field” trials, of each of the insecticides presented in this Manuscript, and (b) the effectiveness of MD-PPF already in field use, whether it is possible to estimate, or envision, the effect of implementing MD with other products (mainly DFB) in routine control.

As with the point of leakage into closed cups above, we feel that it is too early to speculate about the potential of MD-DFB to reach routine control. As we mention in the manuscript (e.g., l. 46, 312-319), we will first need to see if it works in small-scale field trials (with entomological endpoints) – and then, based on those results, decide whether it seems reasonable to run larger-scale trials with epidemiological endpoints. We now stress external validity issues in a new paragraph on study limitations (see l. 294-299).

Minor comments

Table I, legend, “(from disposable lab coats)” – This is already included in the methodology. There’s no need to repeat it

We prefer to keep this small detail so that the legend is self-explanatory.

Discussion, line 258, “DFB (Fig. 2, Table II)” - or Figure 4?

Corrected, thanks.

## Reviewer #3

a) Adequacy of the abstract; b) Originality and importance of the contribution for the development of the field of study. And relevance of: c) Methodology, results and discussion; d) References; e) Figures and tables.

The correspondence we received contained no comments from Reviewer #3.

---

## [Reviewer Report · REVIEWERS COMMENTS]

## Reviewer #1

The authors took all my previous comments into consideration, addressing or justifying each point. The manuscript has improved in consistency and argumentation.

I have only two additional comments:

- in the Introduction, regarding lines 84-85: “(iii) are formulated as powder particles suitable for mosquito-driven dissemination.” In the previous version, I questioned the feasibility of using liquid formulations in dissemination stations. There is a publication in this journal, MIOC, that addresses this topic, and it would be pertinent to consider:

https://www.scielo.br/j/mioc/a/TtV9btYbySrb7yvnDgMmnFc/?lang=en

- in the Discussion, regarding the previous reference to the section “Because we found some evidence suggesting leakage of PPF,” I thank the authors for their response and for including the possibility of leakage also in samples with DFB. Just for clarification, I was comparing, in Figure 4, the closed and open cups of each product tested. In this figure, judging by the confidence intervals presented, there is a difference (closed vs. open) in the three products—even for SPN. Although, evidently, this difference (closed vs. open cups) is much smaller for SPN than for the other two products, the confidence intervals between the two types of cups do not overlap in this experimental condition. It was on this observation that I based my previous comment.

- By the way, still regarding Figure 4, it is worth noting that there is a difference, for PPF, between the value for the closed cups shown in the legend (0.78) and the value in the graph (below 0.75). It is likely that the graph is showing the result including the closed cup that was suspected of contamination (excluded from the value shown in the legend). However, to avoid confusion in interpretation, it might be interesting to make this point in the legend more explicitly, or perhaps (?) to put both results on the graph, considering and not considering the ‘contaminated cup’.

## Reviewer #2

Authors have addressed the concerns point by Reviewers.